# Information, certainty, and learning

**Justin A Harris[1]\*, Charles Randy Gallistel[2]**

[1]The University of Sydney, Camperdown, Australia; [2]Rutgers University, New Brunswick, United States

## eLife Assessment

This paper presents **fundamental** research showing that the acquisition and expression of Pavlovian conditioned responding are lawfully related to temporal characteristics of an animal's conditioning experience. It showcases a rigorous experimental design, several different approaches to data analysis, careful consideration of prior literature, and a thorough introduction. The evidence supporting the conclusions is **compelling**. The paper will have a general appeal to those interested in the behavioral and neural analysis of Pavlovian conditioning.

**\*For correspondence:**
justin.harris@sydney.edu.au

**Competing interest:** The authors declare that no competing interests exist.

**Abstract** More than four decades ago, Gibbon and Balsam (1981) showed that the acquisition of Pavlovian conditioning in pigeons is directly related to the informativeness of the conditioning stimulus (CS) about the unconditioned stimulus (US), where informativeness is defined as the ratio of the US-US interval ($C$) to the CS-US interval ($T$). However, the evidence for this relationship in other species has been equivocal. Here, we describe an experiment that measured the acquisition of appetitive Pavlovian conditioning in 14 groups of rats trained with different $C/T$ ratios (ranging from 1.5 to 300) to establish how learning is related to informativeness. We show that the number of trials required for rats to start responding to the CS is determined by the $C/T$ ratio, and the specific scalar relationship between the rate of learning and informativeness is similar to that previously obtained with pigeons. We also found that the response rate after extended conditioning is strongly related to $T$, with the terminal CS response rate being a scalar function of the CS reinforcement rate ($1/T$). Moreover, this same scalar relationship extended to the rats' response rates during the inter-trial interval, which was directly proportional to the overall rate of reinforcement in the context ($1/C$). The findings establish that animals encode rates of reinforcement, and that conditioning is directly related to how much information the CS provides about the US. The consistency of these observations across species, captured by a simple regression function, suggests a universal model of conditioning.

## Introduction

More than a century of laboratory-based research has been devoted to investigating how animals learn about simple relationships between events, such as learning to respond to a conditioned stimulus (CS) that is followed by an unconditioned stimulus (US) or learning to perform a specific action that is reinforced by a rewarding US. Much of that research has focussed on identifying what properties about the CS-US or response-US relationship are most important for learning. There is widespread consensus about the importance of three particular properties. One is the *temporal contiguity* between the events: conditioning emerges sooner when the US follows the CS or response closely in time. Another is the *spacing* of the learning trials: conditioning takes fewer trials when there is a long time interval between each CS-US or response-US pairing. The third property is the *contingency* between the events: conditioning is more successful when the US occurs reliably in the presence of the CS or response and does not occur in their absence.

**eLife digest** All animals, including humans, can learn when one event signals that another is about to occur, such as when a flash of lightning signals a thunderclap a few seconds later. Such learning is acquired more quickly when the events are closer together in time. However, when events occur less frequently, spacing learning episodes apart is more effective than cramming them together.

Most theories of associative learning assume that it involves the gradual strengthening of a mental connection between representations of the two events each time they are experienced together. This often uses the same simple rules seen in neural networks that power modern AI. However, these theories are not particularly well-suited to explaining why learning is so sensitive to the timing and frequency of learned events. This is why AI finds it easier to learn sequential dependencies in language than to predict events in real time.

Harris and Gallistel sought to determine if they could replicate the decades-old finding that the timing and frequency of events influence learning reciprocally, as they affect how informative the first event is about when the second will occur. Here, informativeness is defined as the time separating each instance of the second event divided by the time between the first and second events. For example, the longer the wait between thunderclaps, and the shorter the wait after each flash of lightning, the more informative the lightning is about the thunder.

Harris and Gallistel trained rats to associate food with a light stimulus using various timing intervals between training trials. The results showed that the speed at which rats learned that a light predicts food can be explained by how informative the light is regarding the presence of food. If the rat knows how infrequent the food is (i.e., how long the rat must wait between food deliveries) and how quickly the food follows the light, the researchers could use the ratio of these intervals to predict how long it takes the rats to learn.

Previous research has shown this in pigeons, but Harris and Gallistel are the first to extend this to a different species, thereby establishing a general principle of learning. Moreover, the frequency with which the rats checked for food was directly proportional to how long they had to wait for it, suggesting the rats learned about the temporal relationship between light and food.

The findings of Harris and Gallistel highlight the differences in how real animals and artificial networks achieve learning. This should guide the efforts of both neuroscientists studying real brains and computer scientists aiming to reproduce animal intelligence.

## Acquisition of conditioning, C-over-T, and informativeness

A landmark study, conducted more than 40 years ago, demonstrated that the first two of these properties—contiguity and trial spacing—are interdependent and subserved by a single principle that encompasses all three properties. In two large-scale experiments, *Gibbon, 1977a* measured the number of trials required for pigeons to start pecking at a key-light (the CS) as they learned it was followed by food (the US). Different groups of pigeons were trained with different trial durations (i.e. the interval between onset of the CS and delivery of the US; henceforth $T$). $T$ ranged from 1 s to 64 s across 41 groups in two experiments. The inter-trial interval (ITI) also varied between groups, ranging from 6 s to 768 s. Gibbon et al. recorded the number of training trials required for each pigeon to start responding reliably during the CS (responding on 3 out of 4 consecutive trials). The birds required more trials as $T$ increased, confirming the effect of CS-US contiguity. They also required fewer trials as the ITI increased, confirming the trial-spacing effect. More importantly, however, these two effects were complementary; an increase in $T$ had no effect if it was accompanied by a proportional increase in the ITI. Thus, the rate at which the pigeons acquired the conditioned response systematically increased as the ratio of ITI to $T$ increased, but there was no separate effect of varying ITI or $T$ when their ratio was held constant.

The extent of the relationship between contiguity and trial spacing was established by a meta-analysis (*Gibbon and Balsam, 1981*) that combined data from all 41 groups tested by *Gibbon et al., 1977b* with data from 11 other experiments investigating the acquisition of key-peck responses in pigeons. This analysis compared the number of trials to criterion against the ratio of $C/T$ (where $C$ is the interval between USs, equal to $T$+ITI). The data on pigeon autoshaping are well described by a regression model whose only parameter is the x-intercept, the $C/T$ value that produces acquisition

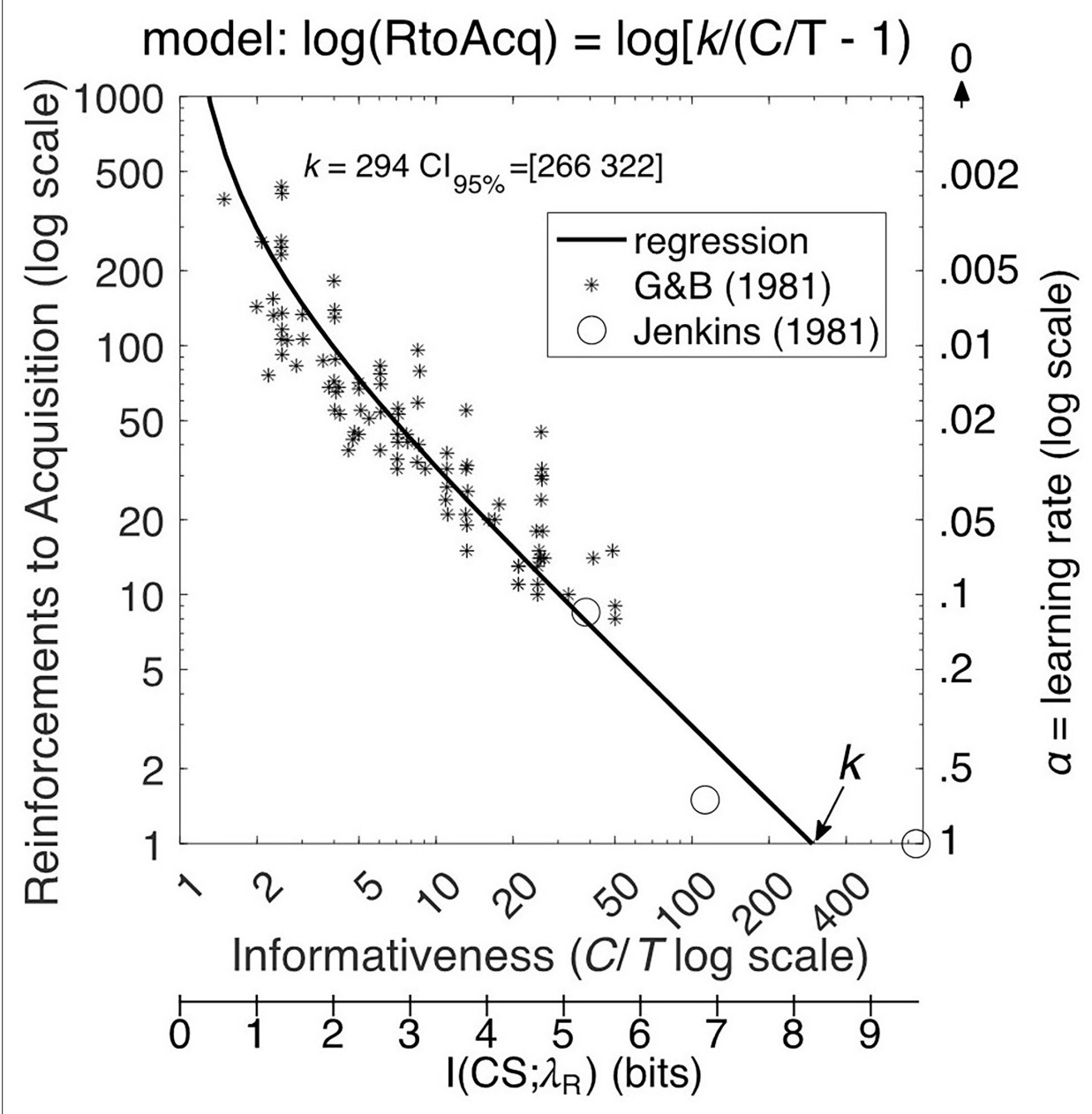

**Figure 1.** Median reinforcements to acquisition plotted against the C/T ratio. Note: Asterisks show the data from *Gibbon and Balsam, 1981* meta-analysis of acquisition in pigeons. The two open circles are from *Jenkins et al., 1981* who used even bigger C/T ratios in some groups. The log of the informativeness is the mutual information, I, between the conditioned stimulus (CS) and the expected wait for reinforcement ($\lambda_R$; lower x-axis). The learning rate (right axis, reversed) is the reciprocal of reinforcements to acquisition (thus the axis increases towards zero). The regression model was fit only to the asterisks, but it also predicts the Jenkins et al. data. The value k is the x-intercept, the informativeness that produces one-trial learning. The regression model curves upward to infinity as informativeness goes to 1 and mutual information to 0 because, when the CS rate does not differ from the contextual rate of reinforcement, differential responding to the CS does not emerge (*Rescorla, 1967*; *Rescorla, 1968*).

after a single trial in the median pigeon (k in *Figure 1*). When C/T>4, as it is in most Pavlovian proto-cols, the slope of the regression model on log-log coordinates is approximately –1. Thus, over most of the C/T range, the number of reinforcements at acquisition is inversely proportional to the C/T ratio; doubling the ratio reduces required reinforcements by a factor of 2.

When reinforcements are delivered only during the CS, the C/T ratio is the ratio of the rate of reinforcement during CSs to the *contextual rate*, the rate subjects expect when in the experimental chamber, without regard to whether the CS is or is not present. *Balsam and Gallistel, 2009* used

the term *informativeness* to refer to the ratio of the CS rate of reinforcement to the contextual rate, because the log of informativeness is the *mutual information* that CS onsets transmit to a subject about the expected wait for the next reinforcement (lower x-axis in *Figure 1*). The mutual information transmitted by a CS is the upper limit on the extent to which the CS can reduce the subject's uncertainty about the wait for the next reinforcement. When the CS rate equals the contextual rate, no information is transmitted because the informativeness ratio is 1, and log(1) = 0.

*Contingency* is mutual information divided by available information (*Gallistel and Latham, 2023*). *Available information* is the amount that reduces a subject's uncertainty to 0. Because temporal measurement error scales with duration measured (Weber's Law, *Gibbon et al., 1977b*), contingency = 1 only when two events coincide in time (*Gallistel and Latham, 2023*).

The analysis presented thus far identifies a fundamental principle of associative learning; what animals learn about events is defined in terms of measurable properties of their temporal distributions. To claim this as a general principle of learning, we must seek evidence for the importance of $C/T$ in conditioning paradigms with other species. Several studies have sought evidence that conditioning is related to $C/T$ using an appetitive Pavlovian conditioning paradigm in which rats or mice learn to anticipate the arrival of a food reward (indexed by monitoring their activity at the food cup or licking at a spout) during a CS (*Bouton and Sunsay, 2003*; *Burke et al., 2024*; *Holland, 2000*; *Kirkpatrick and Church, 2000*; *Lattal, 1999*; *Thrailkill et al., 2020*; *Ward et al., 2012*). However, as summarised below, the evidence from these studies is mixed.

The first evidence for an effect of $C/T$ on the acquisition of responding in rats was reported by *Lattal, 1999* and *Holland, 2000*, who observed effects that persisted even when all rats were tested with an identical ITI (different groups had been trained with different ITIs but were shifted to a common interval between trials when tested for responding to the CS). More recently, *Ward et al., 2012* reported that the log of number of trials for mice to acquire responding scaled with log($C/T$), and this relationship was similar to that described for pigeons (*Gibbon et al., 1977b*; *Gibbon and Balsam, 1981*). *Bouton and Sunsay, 2003* also provided evidence for the importance of $C/T$ on Pavlovian conditioning in rats. They showed that conditioning was negatively affected when $T$ was increased threefold by inserting two CS-alone presentations between each CS-US trial (a manipulation that affects $T$ but does not affect $C$ and, therefore, reduces $C/T$) but that conditioning was not affected by a threefold increase in $T$ brought about by omitting the US from two out of three CS-US trials (a manipulation that increases both $T$ and $C$ equally and, therefore, does not change $C/T$). Most recently, *Burke et al., 2024* have shown that removing 9 out of 10 CS-US trials from a Pavlovian conditioning schedule, thus increasing $C$ 10-fold without changing $T$, reduced the number of trials required for learning by a factor of 10.

In addition to finding an effect of $C/T$ on conditioning, both *Lattal, 1999* and *Holland, 2000* also found an effect of $T$ that was independent of the $C/T$ ratio. They observed that an increase in $T$ resulted in a decrease in responding even when the ITI was also increased to keep the $C/T$ ratio constant. Further evidence against an influence of $C/T$ ratio comes from a study by *Kirkpatrick and Church, 2000* in which differences in $C/T$, ranging from 1.5 to 12, did not produce differences in the acquisition of conditioned responding in rats. Most recently, *Thrailkill et al., 2020* observed clear effects of $T$, but not $C/T$, on conditioning. Their rats responded at much higher rates to a 10 s CS than to a 60 s CS, even though the groups had identical $C/T$ ratios. When comparing groups on how quickly responding was acquired, they found no systematic effect of $C/T$ on the number of 4-trial conditioning blocks required to reach a response criterion.

In sum, in contrast with the impressive evidence from experiments with pigeons, studies of appetitive conditioning in rats and mice have provided inconsistent evidence for the importance of $C/T$ to conditioning. At the same time, these studies have shown that $T$ can affect responding independently of $C/T$. This latter observation is, in fact, consistent with the evidence from experiments with pigeons. In the original study that established the importance of $C/T$ on trials to acquisition, *Gibbon et al., 1977b* also observed that the rate at which the pigeons responded to the CS after extended training was negatively related to $T$ and not related to $C/T$. In other words, while $C/T$ affected how quickly conditioned responding emerged, $T$, rather than $C/T$, determined the level of responding that was ultimately acquired. This distinction may go some way to explaining the inconsistency in the evidence for the effect of $C/T$ in rats and mice. The inconsistencies may be due to differences in how the point of acquisition was identified in different studies, and whether differences in the amount of responding

**Table 1.** Summary of groups.

| Group | T | T range | C | ITI [range] | C/T |
|---|---|---|---|---|---|
| 1 | 42 | 2–82 | 63 | 21 [15-27] | 1.5 |
| 2 | 48 | 2–94 | 144 | 96 [15–177] | 3 |
| 3 | 21 | 2–40 | 94.5 | 73.5 [15-132] | 4.5 |
| 4 | 12 | 2–22 | 72 | 60 [15–105] | 6 |
| 5 | 6 | 2–10 | 54 | 48 [15–81] | 9 |
| 6 | 62 | 2–122 | 930 | 868 [15–1721] | 15 |
| 7 | 30 | 2–58 | 600 | 570 [15–1125] | 20 |
| 8 | 18 | 2–34 | 486 | 468 [15–921] | 27 |
| 9 | 36 | 2–70 | 1296 | 1260 [15–2505] | 36 |
| 10 | 26 | 2–50 | 1404 | 1378 [15–2741] | 54 |
| 11 | 8 | 2–14 | 576 | 568 [15–1121] | 72 |
| 12 | 16 | 2–30 | 1760 | 1744 [15–3473] | 110 |
| 13 | 10 | 2–18 | 1800 | 1790 [15–3565] | 180 |
| 14 | 14 | 2–26 | 4200 | 4186 [15–8357] | 300 |

All values for T (CS-US interval), C (US-US interval), and the inter-trial interval (ITI) are in seconds.

affected the measure of when responding was first acquired. This concern is not relevant to the results from pigeon experiments because the appearance of their key-peck response is unambiguous thanks to the fact that the baseline rate of that response is effectively zero. In contrast, rats and mice show activity at the food cup in the ITI, and therefore researchers using this appetitive paradigm must decide how to take account of the baseline response rate when quantifying conditioned responding during the CS (*Lattal, 1999*).

## The current experiment

The experiment described here attempts to elucidate the role of *C/T* and *T* in an appetitive Pavlovian conditioning paradigm with rats by distinguishing their impact on the emergence of responding from their effect on the level of responding subsequently acquired after extended conditioning. Fourteen groups of rats were trained for 42 sessions with a single CS that was followed on every trial by delivery of food (the US). Each session contained either 10 CS-US trials (Groups 1–11) or 3 CS-US trials (Groups 12–14). Both *T* and *C* varied between the groups in an uncorrelated fashion (*r*=–0.19, *p*=0.519) so that effects of *T* and *C* could be assessed independently (summarised in *Table 1*).

When the location of the CS is spatially separated from the US, as in most appetitive conditioning experiments with pigeons or rodents, two types of response are mutually incompatible, which can impact on the measurement of conditioned responding. For example, any factor that increases food-cup activity, such as reducing the ITI, may reduce evidence for conditioning that is indexed by CS-directed responses like key-pecks. Conversely, evidence for conditioning indexed by food-cup activity may be reduced to the extent that animals also acquire CS-directed responses. These problems can be mitigated if the CS and US are co-located. In the present experiment, the CS was illumination of a small LED inside the magazine. Conditioned responses were measured using an infra-red photo-beam across the opening of the magazine that should detect both food-cup activity and approach responses to the CS. CS-US intervals varied randomly from trial to trial (around a mean equal to *T*) so that response rates remained constant across the length of each trial (see *Appendix 1—figure 1*; and *Harris and Carpenter, 2011*; *Harris et al., 2011*).

## Results

Rats in all groups eventually poked at higher rates during the CS than during the ITI. (See Supplementary materials for mean response rates of each group as well as CS and ITI response rates of each individual rat.) Except for one rat (#104) that registered very few responses across the whole experiment, the difference between CS and ITI response rates over the final five sessions was significant for every individual rat ($p<0.05$, one-tailed Wilcoxon signed rank test) and decisively so in 96% of rats ($p<0.001$). Trial-by-trial response data for every rat are available from https://osf.io/vmwzr/.

### Learning rate (trials to acquisition)

Given that the evidence for response to the CS was decisive in all but one rat, the question of interest is not *whether* there was a significant effect of CS reinforcement on responding, but *when* it first appeared. Our operational definition of learning rate is the reciprocal of reinforcements prior to the appearance of the conditioned response. In our protocol, the conditioned response has appeared when the response rate during the CS is greater than the overall response rate in the experimental context (i.e. the rate across both the ITI and CS intervals, which should not differ before the CS has acquired control over responding). To estimate the point of acquisition in each rat, we calculated the difference between the cumulative response rate during the CS at each trial (cumulative number of responses during the CS divided by the cumulative duration of the CS) and the cumulative contextual response rate (across the CS and ITI). We took the trial when this difference became permanently positive as the first trial on which the rat had acquired responding to the CS (i.e. when the cumulative CS response rate was permanently greater than the cumulative contextual rate). Therefore, the trial before this was our first estimate of the number of reinforcements to learn. *Figure 2A* plots the estimated reinforcements to acquisition for each individual rat and the median for each group. It shows a steady decrease in the number of reinforced trials prior to acquisition as the *C/T* ratio increased, thus confirming the scalar relationship between learning rate and informativeness. The data are provided in Acquisition_Table.xlsx available at https://osf.io/vmwzr/.

The data in *Figure 2A* put a number on trials to acquisition for each rat, but these estimates come without an index of the strength of evidence, such as an odds ratio or *p*-value. We have used an information-theoretic statistic, the $nD_{KL}$ (*Gallistel and Latham, 2023*), to evaluate the strength of evidence at each trial that the rat's CS response rate exceeds its overall response rate. The $nD_{KL}$ estimates the cumulative coding inefficiency of a model that treats the response rate during the CS as indistinguishable from the contextual response rate. An important advantage of the $nD_{KL}$ is that, unlike a Fisherian *p*-value, its interpretation does not require specification of sample size in advance because the effective sample size <u>is</u> the *n* in the $nD_{KL}$. This means that the test can be applied iteratively with the data from each successive trial, where *n* is the number of trials up to each iteration. Another useful property of the $nD_{KL}$ is that its value, measured in nats (natural-log units), can easily be converted into more familiar odds ratios or *p* values. A detailed explanation of the $nD_{KL}$ and justification for its use are provided in the Methods section at the end of this paper.

We iteratively computed the $nD_{KL}$ for the difference between CS responding and context responding across trials. Because divergence is a scalar measure of distance, rather than a measure of difference, the $nD_{KL}$ is positive even when the rate of responding during the CS is less than the overall response rate. In our experiment, some rats did initially respond less during the CS, which we put down to a temporary disruption to ongoing magazine activity by the unfamiliar magazine light. Therefore, because we are only interested in evidence for a positive difference between CS and context response rates, we have added a negative sign to the $nD_{KL}$ on trials where the cumulative CS rate is less than the overall rate. As a result, the trial on which the $nD_{KL}$ becomes permanently positive is the trial when the cumulative CS rate permanently exceeds the overall rate (as plotted in *Figure 2A*). The trial-by-trial values of the $nD_{KL}$ for every rat are plotted in the figures for individual rats in the Supplementary Materials and are included in the data file nHT.mat available at https://osf.io/vmwzr/. For rats that responded less during the CS, such that their $nD_{KL}$ was initially negative (e.g. Rat #10), the $nD_{KL}$ began to increase from its lowest point when responding during the CS began to increase, yet it could take many trials before the $nD_{KL}$ became positive. In these cases, a more accurate marker of acquisition of responding to the CS would be the minimum of the $nD_{KL}$ rather than the trial on which it became permanently positive. However, the minimum $nD_{KL}$ is a questionable localiser of acquisition in other cases where it does not mark the start of a systematic increase in the $nD_{KL}$ (e.g. Rats 1 and 7).

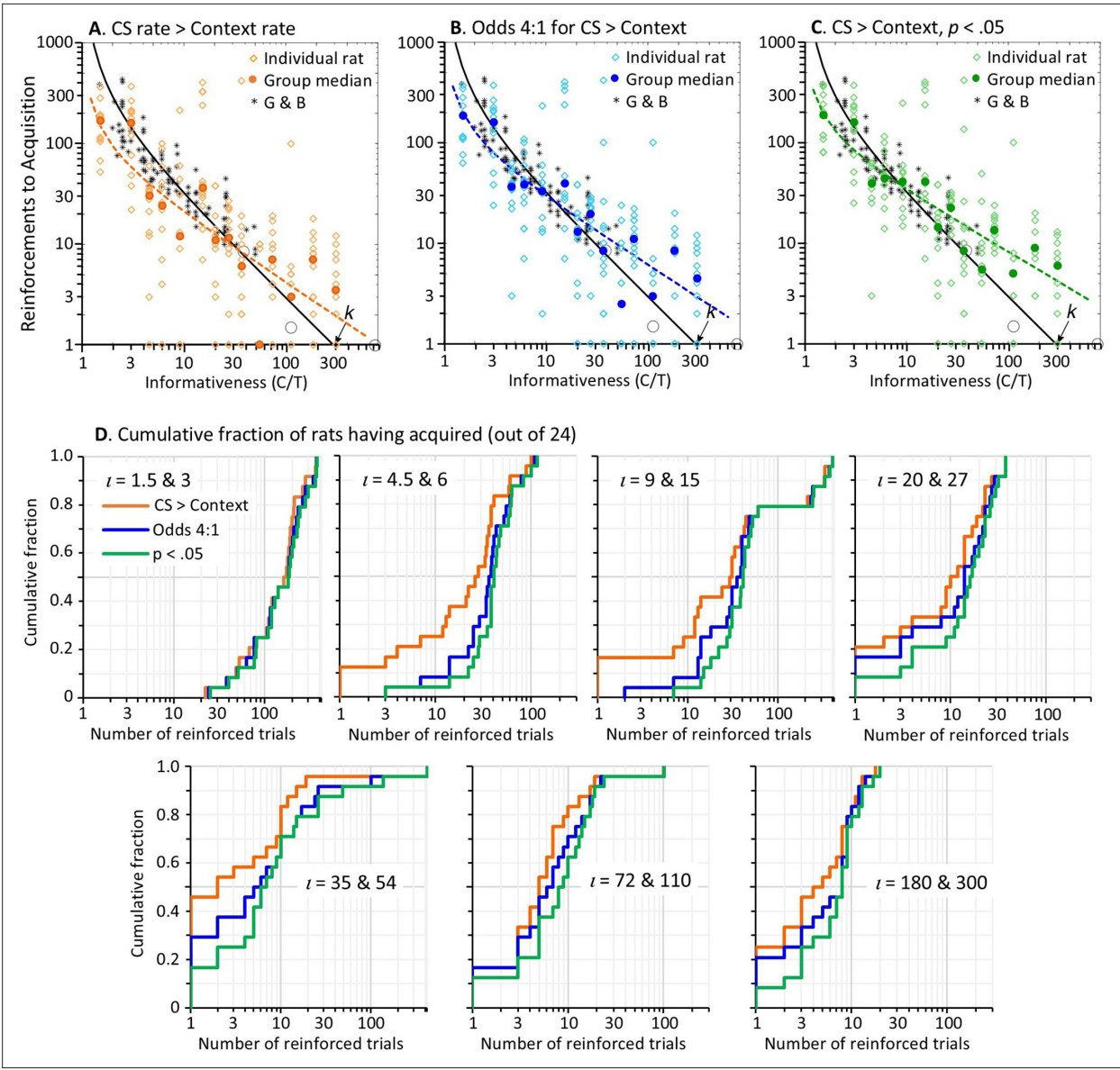

**Figure 2.** Number of trials for acquisition of responding to the conditioned stimulus (CS), plotted against CS informativeness (*C/T* ratio). Note. A, B, and C plot the number of trials to reach three different acquisition criteria for CS response rates exceeding the contextual rate. Each plot shows the median number of trials to criterion for each group (filled coloured circles) and the number of trials for each rat (unfilled coloured diamonds) plotted against informativeness of the CS. These data are superimposed onto the data (black asterisks) from **Gibbon and Balsam, 1981** and the black regression line shown in **Figure 1**. The dashed coloured lines show the regression model, log[RtoAcq]=log[k·(C/T-1)$^s$], fitted to the group medians. The slope, s, and R$^2$ of each regression line is: in A, s=−0.68 [95% confidence intervals = −0.94, −0.42], R$^2$=0.73; in B, s=−0.66 [-0.86, −0.46], R$^2$ = 0.81; in C, s = −0.6 [–0.76, –0.45], R$^2$=0.86. The seven plots in D show the cumulative fraction of rats that had acquired responding to the CS after n reinforced trials (x-axis) according to the same three acquisition criteria in A to C. In each plot, the data from two groups with similar informativeness (ι, iota) are combined to increase the sample size.

Nonetheless, we have included the minimum $n\text{D}_{\text{KL}}$ as an index of acquisition in the data plots for each rat in the Supplementary Materials and in Acquisition_Table.xlsx.

We have used the $n\text{D}_{\text{KL}}$ to identify in each rat when the evidence for acquisition, based on a positive difference between its response during the CS and its overall response rate, had exceeded one of two statistical thresholds. The more lenient threshold was for odds of at least 4:1 that CS responding was higher than the overall response rate; this threshold is met when the $n\text{D}_{\text{KL}}$ is permanently greater than 0.82 (**Gallistel and Latham, 2023**). While odds of 4:1 would normally be regarded as only weak evidence for an effect, it is important to remember that the evidence for acquisition was eventually

decisive in every subject. Our second, more stringent, criterion was set at $p<0.05$ (odds 19:1) for the probability that the CS response rate was not more than the overall rate; this threshold is met when $nD_{KL} >1.92$. For both of these statistical criteria, we calculated the $nD_{KL}$ starting from the trial on which the cumulative CS response rate was permanently greater than the contextual response rate. We started from this trial, rather than from Trial 1, because response rate data from trials prior to the point of acquisition would dilute the evidence for a statistically significant difference in responding once it had emerged, and thereby increase the number of trials required to observe significant responding to the CS. The data from Rat 1 illustrates this point. The CS response rate of Rat 1 permanently exceeded its overall response rate on Trial 52 (when the $nD_{KL}$ became permanently positive). The $nD_{KL}$, calculated from that trial onwards, surpassed 0.82 (odds 4:1) after a further 11 trials (on Trial 63) and reached 1.92 ($P<.05$) on Trial 81. By contrast, the $nD_{KL}$ for this rat, calculated from Trial 1, did not permanently exceed 0.82 until Trial 83 and did not exceed 1.92 until Trial 93, adding 10 or 20 trials to the point of acquisition.

*Figure 2B* plots the number of trials for each rat to reach odds greater than 4:1 of a difference between CS responding and overall responding. *Figure 2C* plots the number of trials to reach $p<0.05$ against that difference in response rates. (Other trials to criterion using statistical thresholds $p<0.01$ and.001 are included in Acquisition_Table.xlsx.) As expected, the trials to acquisition in *Figure 2B and C* are higher than those shown in *Figure 2A* simply because it requires additional trials to accumulate statistical evidence of the difference between CS responding and overall responding. Nonetheless, in all three plots, there is clear evidence that the log of trials to acquisition scales with the log of CS informativeness, a fact confirmed by regression lines plotted to each set of data based on the regression model presented in *Figure 1*. Indeed, the data in *Figure 2* are superimposed on *Figure 1*, revealing a striking correspondence between the data from *Gibbon and Balsam, 1981* meta-analysis of learning rates in pigeons and our data with rats, especially in *Figure 2B and C* where statistical thresholds have been applied.

To assess how specific the relationship is between trials to acquisition and informativeness, we performed correlational analyses for the relationship between the median log of trials to acquisition and log($C/T$), log($C$), and log($T$). We set α at 0.017 to correct for multiple tests. Log($C/T$) was the strongest predictor of trials to criterion ($r=-0.84$, $-0.89$, and $-0.92$; $p$'s$<0.001$; for the three criteria shown in *Figure 2A, B and C*, respectively). Log($C$) was also significantly correlated with the log of trials to criterion ($r=-0.71$, $-0.81$, and $-0.83$; $p$'s$<0.006$) but log($T$) was not significantly correlated with trials to any criterion ($r=0.43$, $0.35$, $0.35$; $p=0.121$, $0.221$, and $0.225$). Given that log($C$) and log($C/T$) were strongly correlated with one another ($r=0.90$), we calculated their partial correlations to test whether each made independent contributions to the rate of acquisition. After partialling out the effect of log($C$), correlations with log($C/T$) remained significant ($r$'s$=-0.65$, $-0.64$, and $-0.69$; $p\leq0.017$). In contrast, after partialling out the effect of log($C/T$), each correlation with log($C$) was not significant ($r$'s$=0.19$, $-0.03$, and $-0.07$; $p$'s$=0.543$, $0.916$, and $0.816$). In sum, the number of trials required for conditioned responding to emerge was strongly affected by the $C/T$ ratio, whereas neither $C$ nor $T$ alone had any independent effect.

From the plots in *Figure 2A, B and C*, it can be hard to appreciate the distribution of trials to acquisition when the data for individual subjects are superimposed. This is particularly the case for subjects learning after their first reinforcement, where individual data pile up on the x-axis. To show the distribution more clearly, *Figure 2D* contains seven plots of the cumulative fraction of animals that had acquired responding to the CS by trial $n$ (x-axis) based on the three criteria shown in *Figure 2A* to C. To increase the sample size, each plot combines data from two groups with similar informativeness ratios ($\iota$, iota). Three things become particularly evident in these plots. One is that the curves migrate leftwards as informativeness increases, showing that the whole distribution of subjects takes fewer trials to reach each criterion for learning. The second observation is that, apart from the two groups with the lowest informativeness ($\iota=1.5$ & 3), many rats (33 out of 144, 23%) acquired responding in just 1 trial, as determined by the criterion that the CS response rate permanently exceeded the contextual rate (orange lines). Of these 33 rats, the evidence for 1-trial learning exceeded 4:1 odds in 60% of cases and was significant at $p<0.05$ in 30% of cases. The third observation is that, while the number of rats showing 1-trial learning increased as informativeness increased from low to intermediate levels, 1-trial learning was less common among rats trained with the highest levels of informativeness. This is surprising because 1-trial learning

should be most frequent in rats trained with the highest informativeness. We will return to this issue shortly.

The regression lines in *Figure 2A, B and C* confirm *Gibbon and Balsam, 1981* observation of timescale invariance in the rate of learning, in that trials-to-acquisition scale with the ratio of $C$ over $T$. Indeed, the rates of learning in the first 10 groups of rats ($C/T$ ratios <72) agree quite closely with those reported by Gibbon and Balsam across a comparable range of $C/T$ ratios, particularly when a statistical threshold was applied to the evidence for responding (panels B and C). However, the slopes of the regressions fitted to the data from all 14 groups are less negative than the slope (–1) of the black regression line modelled on Gibbon and Balsam's data. The primary reason for the shallower gradients in our regressions appears to be the higher-than-expected number of trials to acquisition for the four groups trained with the highest $C/T$ ratios (≥72). Indeed, if we fit regressions to our data but excluding those four groups, thereby restricting the analysis to the range of informativeness values used by Gibbon and Balsam, we obtain slopes closer to –1: for *Figure 2A*, slope = −0.93 [95% confidence intervals = −1.33, –0.54]; for *Figure 2B*, slope = −0.82 [-1.11, –0.52]; for *Figure 2C*, slope = –0.75 [–0.97, –0.50]. This reiterates the surprising observation made above that the number of rats showing 1-trial learning was smaller in these four groups than in groups trained with intermediate informativeness. It seems implausible that a rat should learn more from one exposure to a moderately informative CS than to a CS that is six times more informative about the US.

The fact that groups trained with very high informativeness acquired more slowly than expected, and were less likely to show acquisition after 1-trial, might imply that the scaling between learning rate and informativeness changes at high levels of informativeness. However, the apparent limit on learning rate at high informativeness is likely to be, at least in part, an artefact of our training protocol and, in particular, to the fact that our rats were not given magazine training—sessions in which food was delivered without the CS—before conditioning with the CS. Magazine training ensures that the animals are familiar with the arrival of food into the magazine and, therefore, can promptly detect it during subsequent conditioning trials with the CS. Without magazine training, it is likely that many rats did not find the food immediately after its delivery on the first few conditioning trials. We have confirmed this in a recent unpublished experiment that delivered food pellets at widely spaced intervals (on average every 45 min) and measured the latency for rats to enter the magazine after each pellet was delivered. The median latency for the 32 rats to enter the magazine was 39 s on Trial 1, which decreased to 2 s after five trials. Such a delay in finding the food US would substantially increase the effective CS-US interval, measured from CS onset to discovery of the food pellet by the rat, making the CS much less informative over those initial trials. Crucially, this would affect groups trained with high informativeness much more than those with low informativeness, first because those initial few trials are more critical for establishing learning when informativeness is high, and second because the effective increase in CS-US interval will reduce high levels of informativeness more than low informativeness. To illustrate this second point, consider the impact of a 39 s delay in finding the food pellet for rats in Group 1, with $C$=63 and $T$=42, versus rats in Group 14, with $C$=4200 and $T$=14. For Group 1, the 39 s delay would reduce $C/T$ by just 16%, from 1.5 to 1.26 (=102/81). For Group 14, the same 39 s delay would reduce $C/T$ by 73%, from 300 to 80 (=4239/53). Future experiments could reduce the likelihood of large delays in how quickly the rats find the food by giving them magazine training with the same US-US interval to be used during conditioning, thereby retaining the contextual rate of reinforcement at $1/C$.

Another factor that might have concealed evidence for 1-trial learning in the groups trained with extremely informative protocols is that their initial response rates were so low that, in many cases, the $n\text{D}_{\text{KL}}$ could not be computed for several trials. Take, for example, Rat 176, trained with the highest level of informativeness (300). This rat had been in the chamber for over an hour when the CS first came on for just 4 s. It was in the chamber for another hour before the CS came on again for 15 s. Because the rat did not register a response during either CS presentation, nor during either 30 s pre-CS interval, the $n\text{D}_{\text{KL}}$ could not be computed until Trial 3 when the rat made its first response during the 23 s CS (still failing to register any response in the pre-CS period). At this point, the evidence of responding to the CS was already significant at $p$<0.05 ($n\text{D}_{\text{KL}}$ = 2.35). The complete absence of responses during the first two trials makes it difficult to know whether the rat had in fact learned about the CS on Trial 1 but it failed to register a response during the 15 s presentation of the CS on Trial 2 because its response rate was still very low.

We have developed a principled solution to the problem caused by the lack of responses during the first few short CSs in groups with extreme values of informativeness. This solution uses the $nD_{KL}$ to parse the trial-by-trial response rates into segments of one or more trials based on inter-poke intervals. Response rates for trials within a segment are treated as the same but the boundary between segments marks a significant change in rate. These parsed response rates during the CS and during the ITI are included in the individual data plots for each rat in the Supplementary materials and are in ParseTable.xlsx available at https://osf.io/vmwzr/. When there is a long wait to the first response, followed by a sequence of much shorter inter-response intervals, the parsing algorithm (see Methods) finds a change in rate at that first response, ascribing a response rate of zero before the first response and a higher poke rate in the segment that includes the first response. It does this if the long interval before the first response is unlikely to have come from the distribution of inter-response intervals observed thereafter. However, when the duration of the interval before the first response plausibly belongs to the distribution of subsequent inter-response intervals, the algorithm includes that initial interval within the segment of subsequent responses, such that the onset of the estimated response rate extends back to the start of observation. This can be seen in the parsed data for Rat 176 (see Supplementary materials) which ascribes a single response rate to the CS and another, lower, rate to the ITI across all of the first 10 trials. Acquisition_Table.xlsx includes estimates, based on the $nD_{KL}$, of the number of trials before the parsed CS response rates exceeded the parsed context response rates using increasingly stringent statistical thresholds (odds of 4, 10, 20, 100, & 1000:1 against the null hypothesis that CS response rate equals the context response rate). It shows that, using this method, evidence for 1-trial acquisition with odds >4:1 is obtained in the majority of rats (56%) that were trained with informativeness ratios of 72 and above, and 44% of rats show 1-trial acquisition with odds >10:1.

## Rate of responding as a function of reinforcement rate

After the initial point at which each rat had begun responding to the CS, response rates typically increased as conditioning continued, but there were large differences in how much responding increased. By the end of the experiment, response rates varied by more than three orders of magnitude, from as low as once every several minutes to as high as several times per second. From our analyses (see Supplementary Materials), we realised that the conventional way to compute response rates seriously underestimates the high poke rates observed during some CSs. The conventional calculation of response rate simply divides the poke count by the time interval across which pokes are counted. This implicitly assumes that the durations of the pokes themselves—during which another response cannot be initiated—constitute a negligible fraction of the duration in the denominator. However, this is far from true when there is more than one poke per second. Our analysis showed that the mean duration of a nose poke is 0.5 s, which will curtail the number of responses that can be produced within a fixed interval. Therefore, in our further analyses of response rates, we have corrected the duration in the denominator by subtracting cumulative poke duration from cumulative time to obtain the cumulative interval over which it was possible for the rat to initiate a poke. This is equivalent to computing the rate as the reciprocal of the inter-response interval, measured from the end of one response to the start of the next, except that it can be computed even when there are fewer than two responses in a trial (unlike the inter-response interval). When calculating the ITI response rate over the final 5 (or 10) training sessions, we assigned a pre-CS poke count of 0.5 in cases where the rat did not register a single pre-CS poke (i.e. the recorded count was 0). This served as an unbiased estimate of the true ITI response rate (which would be between 0 and 1/the cumulative pre-CS time) and allowed us to calculate the log of the ITI response rate (shown in *Figure 3* and used to calculate regressions), which is otherwise undefined when the value equals 0.

We have adopted a further correction to the estimate of response rates during the CS. This correction was warranted because the distribution of latencies to the first poke after CS onset is different from the distribution from the intervals separating subsequent pokes (see page 5 of Supplementary materials). This was particularly true during CSs that elicited high response rates, where the inter-poke intervals were often measured in milliseconds, whereas the latencies of the first pokes were usually around 2 s. This latency may reflect a lower limit on how quickly the rats could reach the magazine after CS onset or occur because 2 s was the lower limit on reinforcement latency after CS onset. Regardless of the reason, in our further analyses of CS response rates, we did not include the first

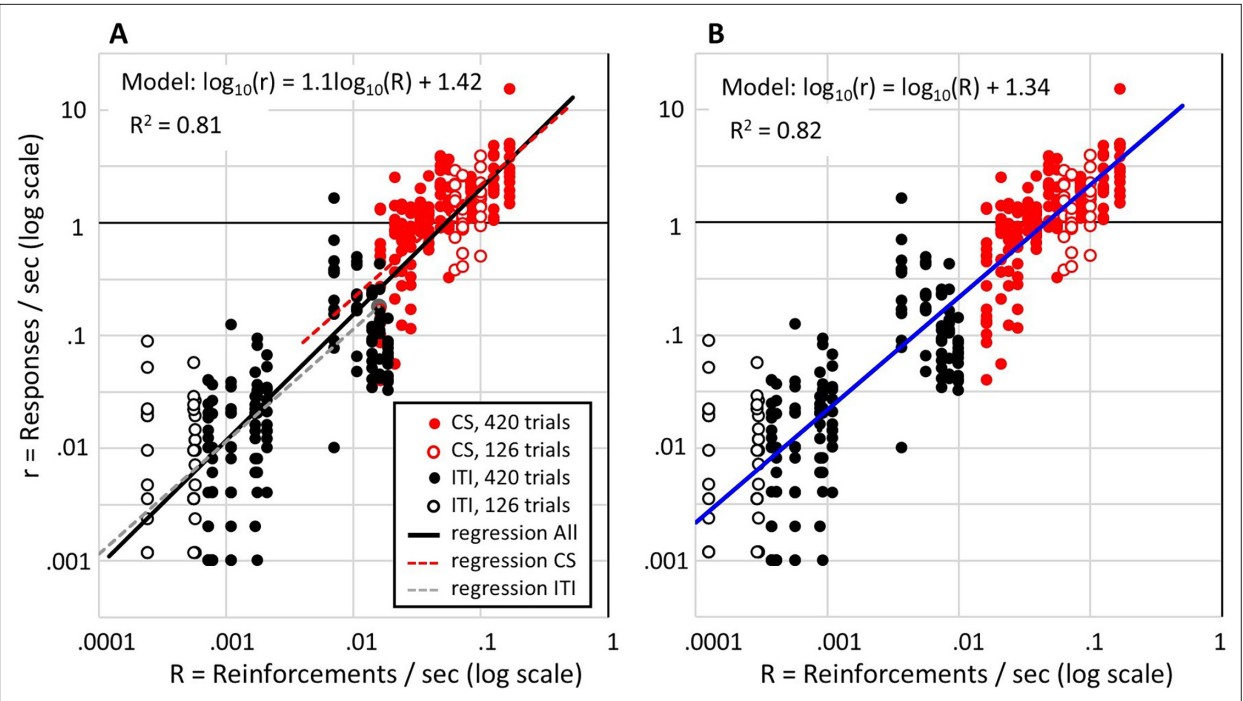

**Figure 3.** Terminal response rates (final five sessions) of individual rats as a function of the reinforcement rate on double-logarithmic coordinates. Note. Red circles show response rates during the conditioned stimulus (CS) (number of pokes after the first poke divided by the remaining CS time when the rat's head was out of the magazine) plotted against the CS reinforcement rate (1 /T). Black circles show response rates during the inter-trial interval plotted against the baseline reinforcement rate (1 /C). Filled circles show data of rats given 420 reinforced trials (Groups with C/T ratios ≤72) and open circles show data of rats given 126 reinforced trials (Groups with C/T ratios ≥110). The model in A gives the equation for the solid black regression line ($R^2$=0.81). The slope of this line is virtually indistinguishable from 1 ($R^2$=0.80, when slope fixed at 1). The dashed red and grey lines show regression lines, with slopes fixed at 1, fitted separately to the CS response rates and inter-trial interval (ITI) response rates. In B, the blue regression line has a slope fixed at 1, and the ITI response rates (black circles) are plotted against the overall reinforcement rate (1 /C) divided by 1.9.

pokes in the poke count, and we subtracted the latency of each first poke from the CS durations. The code used to extract the response rates over the last 5 (or 10) sessions is available at https://osf.io/vmwzr/.

Our analysis of response rates across the last five sessions uncovered several important findings (very similar results are obtained when using data from the final 10 sessions). First, the response rate during the CS scaled with the reinforcement rate of the CS. This is shown in *Figure 3A*, where each red dot shows one rat's response rate during the CS plotted against the CS reinforcement rate (1 /T) for that rat. To extend the evidence for this relationship between response rate and reinforcement rate, the black dots in *Figure 3A* show each rat's response rate during the ITI plotted against the overall reinforcement rate (1 /C) for that rat. This reveals that the ITI response rate scales with the overall reinforcement rate and, strikingly, the scaling is similar to that observed for the CS response rate and reinforcement rate. Indeed, a single regression line (solid black line in *Figure 3A*) plotted through all the data accounts for 81% of the variance in log response rates. It is noteworthy that this regression has a slope very close to 1, and fixing the slope at 1 produces only a small loss of explanatory power ($R^2$=0.80). Fixing the slope at 1 is theoretically justified because it means that, in the non-log domain, the scalar relation between response rate and reinforcement rate passes through the origin. Put simply, it means that the response rate is zero when the reinforcement rate is zero, and the response rate increases in fixed proportion to the reinforcement rate. It is particularly impressive that this simple relation holds across more than two orders of magnitude variation in reinforcement rate, from ITI rates as low as one pellet every 70 min to CS rates as high as 10 pellets per minute. As is also apparent in *Figure 3A*, the variability about the scalar relation between response rate and reinforcement rate also scales with reinforcement rate. That is why the regression must be computed in the logarithmic domain.

*Figure 3A* also includes separate regression lines fitted to the CS data and ITI data, shown as red and grey dashed lines. For both regressions, the slope was fixed at 1 for the reasons just described. (When regressions are fitted to the CS and ITI response rates, with free parameters for the slopes, the slopes are close to 1: for CS rates, slope = 0.91 [95% confidence intervals = 0.75, 1.07]; for the ITI rates, slope = 0.84 [0.71, 0.96]. Using Bayesian Information Criterion scores to compare this 4-parameter model against the 2-parameter model with a fixed slope of 1 favours the 2-parameter model: BIC = -591 vs -589, for 2-parameter and 4-parameter models, respectively.) This 2-parameter model provides a small increase in explained variance ($R^2$=0.82) over the single regression with slope of 1, and the difference in their Bayesian Information Criteria ($-591$ vs $-561$) provides justification for the additional parameter in the separate regressions model. The difference in intercept on the log x-axis for the CS and ITI regressions (difference = 0.28) means a difference of 1.9 in the slopes of the CS and ITI regressions in the non-log domain. In other words, the scaling between reinforcement rate and response rate for the CS is approximately double that for the ITI response rate against 1 /C. However, describing it in these terms assumes that the rat expects reinforcement at a rate of 1 /C during the ITI even though it was never reinforced in the ITI. An alternative, and more likely, possibility is that the rate of responding in the ITI is a function of the rate of reinforcement that the rat ascribed to the context, and this is less than 1 /C because the presence of the CS reduces (overshadows) what is learned about the context's reinforcement rate. Indeed, the difference in slopes for the CS and ITI regressions would suggest that the reinforcement rate ascribed to the context is approximately half the overall reinforcement rate. To illustrate this, *Figure 3B* plots a regression model (blue line) that assumes a single scaling factor between reinforcement rate and response rate across both ITI and CS, but additionally assumes a uniform shift in the reinforcement rate expected during the ITI. That shift, equal to 0.28 log units, is the difference in x-intercepts for the CS and ITI regressions in *Figure 3A*, and represents a uniform rescaling of the expected reinforcement rate by a constant proportion of 1 /C. According to the blue line in *Figure 3B*, the response rate is 21.8 times the reinforcement rate and the reinforcement rate expected during the ITI equals the overall reinforcement rate scaled by approximately 0.5 (=1/1.9).

## Trials from onset of responding to peak responding

Having established the relationship between response rate and reinforcement rate, we next analysed how response rate increased over trials towards its maximum value. Our first analysis assumes that there is a consistent (monotonic) increase in response rate starting from the initial point of acquisition. This analysis followed a method recently described (*Harris, 2022*) that uses the slope of the cumulative response rate over trials to identify the trial on which the response rate had reached each decile (from 10 to 90%) of the peak response rate. Based on our earlier analysis (see *Figure 3*), our measure of the CS response rate was calculated by dividing the response count (excluding the first response in each CS presentation) by the total time out of the magazine during the CS (and excluding the latency to the first response). As shown in *Appendix 1—figure 3A*, the response rate of an individual rat varies greatly from trial to trial. However, a clearer picture of the overall change in responding over trials can be obtained by plotting the cumulative response count against the cumulative opportunity to respond (cumulative time out of the magazine; *Appendix 1—figure 3*).

The slope of the cumulative function can be used to estimate the rat's response rate across conditioning to find when responding had reached a given proportion of the peak response rate. To analyse how response rates changed across the course of conditioning, we extracted a segment of each rat's conditioning data starting from the trial, $t_1$, on which the response rate during the CS became reliably greater than the ITI response rate and finishing at the trial, $t_{end}$, on which the response rate reached its peak (according to a moving average with a window width of three sessions). The total change in responding across conditioning was calculated by subtracting the response rate at the start of this segment of trials, $R_1$, from the peak response rate, $R_{max}$ (at the end of the segment): $\Delta R = R_{max} - R_1$. To identify when responding had increased by 10% of $\Delta R$, we estimated what the cumulative response count, $cumR'$, would be at each trial, $t$, if the rat maintained a fixed level of responding equal to the starting rate plus 10% of $\Delta R$. Thus, $cumR'_t = R_1 + 0.1 \cdot \Delta R \cdot cumT_t$, where $cumT_t$ is the cumulative CS-US interval from trial 1 to $t$. We then calculated the difference between $cumR'_t$ and the observed $cumR_t$. The trial at which this difference was maximum was identified as the trial when the slope of $cumR_t$ had increased by at least 10% of $\Delta R$. This process was repeated for all deciles up to 90%. An example of

the values obtained for one rat is shown in *Appendix 1—figure 3C*. In this example, the rats' response rate had increased by 10% of ΔR on Trial 34, by 50% of ΔR by Trial 147, and by 90% of ΔR by Trial 233.

The analysis illustrated in *Appendix 1—figure 3* was conducted on the individual data of all rats (except those rats missing data from Session 1). We excluded rats with a ΔR less than 0.1 responses/s. The mean number of trials to reach each decile for every rat in the 14 groups is shown in *Figure 4*. With some exceptions, for most rats, the relationship between the number of trials and response decile was roughly linear, meaning that their response rate increased uniformly over trials as it approached the peak response rate. This is clearest in the averaged functions (thick black lines in *Figure 4*). To investigate more precisely the relationship between trials to criterion, $t_c$, and response decile, $d$, we compared 4 different functions for their fit to the data for each individual rat. Based on the apparent linear increase in trials across deciles, the first function tested was a straight line, $t_c = m.d+c$. The second was an exponential function, $t_c = c.e^{m.d}$, which has a continuously increasing slope and thus predicts a systematic increase across deciles in the number of trials between deciles. This function had most successfully captured the relationship between trials and response deciles in the data analysed by *Harris, 2022*. The third function was an inverse cumulative Gaussian: $t_c = s.2^{½}.erf^{-1}[2.(m.d+0.5)–1]+c$. This was used to model a stepwise increase in responding, as would be observed if responding were governed by a decision process when evidence for the CS-US relationship exceeded some threshold (*Gallistel et al., 2004*). The fourth function was a log function, $t_c = -(log_e[1–d])/k+c$, derived as the inverse of a cumulative exponential function. This function models the relationship between trials and response criterion predicted by an error-correction learning algorithm, such as used by the Rescorla-Wagner model (*Rescorla and Wagner, 1972*). All model fitting was conducted using MATLAB. To compare between these four models of the data, the Bayesian Information Criterion (BIC) was calculated as

$$BIC = n \cdot log_e \left( \frac{RSS}{n} \right) + p \cdot log_e n$$

where RSS is the residual sum of squares for the difference between each observed $t_c$ and its corresponding point on the fitted function, $n$ is the number of points being fitted (=9), and $p$ is the number of free parameters in the function.

The overall performance of the functions can be compared by summing the BICs obtained from all rats. The function with the smallest $\Sigma$BIC is the function with the most evidence. According to this analysis, the linear function had the most evidence, $\Sigma$BIC = 9588, followed by the exponential function, $\Sigma$BIC = 9974. The difference between these (ΔBIC = 386) constitutes overwhelming evidence in favour of the linear model: BF = $e\Delta^{BIC/2}$ = 7.3×10$^{83}$ (*Wagenmakers, 2007*). The $\Sigma$BIC of the other two models were much higher again, 12,128 and 10,358, indicating that they had even less support from the data. In addition to this comparison of the aggregate BIC, a more specific comparison can be made by comparing the BIC for one model against the BIC for another for each rat. The scatter plot in the bottom right corner of *Figure 4* plots, for each rat, the BIC for the exponential function against the BIC for the linear function (the two best-fitting functions). The orange line marks where these BICs are equal. The large majority of values (72%) sit above this line. These represent cases where the BIC for the linear function is lower than that for the exponential function, meaning that the evidence is stronger for the linear function. If we look at cases where the difference in BICs was greater than 4.6, corresponding to strong evidence in favour of one function over the other (a BF ≥10), there is strong evidence favouring the linear function over the exponential in 39% of cases, whereas only 9% of cases provide strong evidence in favour of the exponential function. The evidence favouring the linear function over either the inverse cumulative Gaussian or the log function is even stronger: 69% of cases provide stronger evidence (BF ≥10) in favour of the linear over the inverse cumulative Gaussian and 0% of cases favour the latter function; 59% of cases provide strong evidence in favour of the linear over the log function and only 8% of cases strongly favour the log function over the linear.

The above analyses reveal an overall tendency for the response rate to increase approximately linearly up to the point where the peak response rate is reached. To test how the rate of this increase varied across groups, we calculated the correlation coefficient between the slope of the line for each group, as shown in *Figure 4*, and the value of C, T, or C/T. The slope was negatively correlated with C and C/T, $r$s = –0.66 and –.67, $p$s ≤ 0.010, and not correlated with T, $r$=–0.05, $p$=0.864. The correlations with C and C/T were substantially driven by the fact that the three groups with the largest

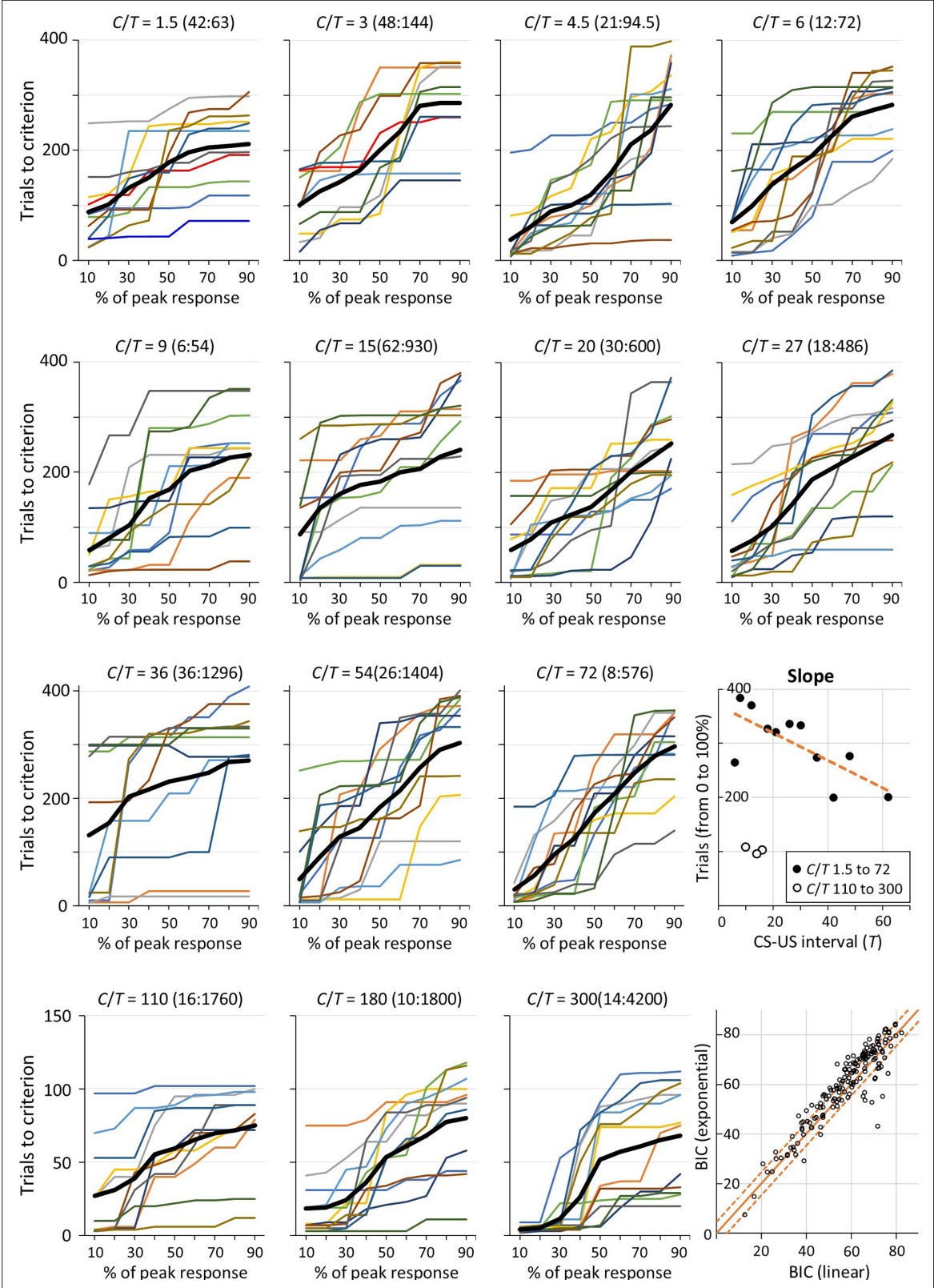

**Figure 4.** Number of trials to each decile of peak response rate. Note: The coloured line plots show the mean number of trials to each decile for each rat in each of the 14 groups (identified by the C/T ratio). The thick black line in each plot is the average for that group. The slopes of those black lines for each group are plotted against each group's T in the rightmost plot of the third row. The final plot in the bottom right corner of the figure plots the Bayesian Information Criterion (BIC) for an exponential function against the BIC for a straight line when each was fitted to the trials-to-criterion

*Figure 4 continued on next page*

*Figure 4 continued*

data of individual rats (each open black circle shows the BICs for one rat). For each point in the top left of the plot (above the solid orange line), $BIC_{line}$ <$BIC_{exponential}$, meaning that the data were better accounted for by the line than the exponential function (and vice versa for points in the bottom right half of the plot). The two dashed orange lines mark where the difference in BICs equals 4.6 which equates to odds of 10:1 in favour of the function with the lower BIC.

values of $C$ and $C/T$ had much smaller slope values than the other 11 groups (slopes <110 vs slopes ≥ 200). It is likely that this difference in slopes was an artefact of the smaller number of trials given to those three groups (126 vs 420 trials) which limited the range of values that their slopes could take. In light of this, we calculated correlations after excluding the three groups given only 126 trials. For the other 11 groups, the slope was not significantly correlated with $C/T$, $r=0.51$, $p=0.110$, or with $C$, $r=0.08$, $p=0.938$, but was significantly negatively correlated with $T$, $r=–0.73$, $p=0.011$ (see plot titled 'Slope' in *Figure 4*). The correlation between slope and $T$ remained significant after partialling out the effect of $C/T$, $r=–0.70$, $p=0.025$. This suggests that the response rate increased more quickly when the reinforcement rate of the CS ($1/T$) was higher, which is not surprising given that the peak rate of responding also increased systematically with reinforcement rate (*Figure 3*).

The preceding analysis assumes that there is an overall monotonic increase in responding over trials. However, this trend is not always apparent, particularly when the CS informativeness is low. Such irregular and non-monotonic changes in responding are revealed by the analysis that uses the $nD_{KL}$ to parse the response rates into segments that have significantly different rates (either higher or lower). (See description of response rate parsing using the Kullback-Leibler divergence and the $nD_{KL}$ in Methods.) The parsed response rates are shown in the bottom two plots of the individual figures for each rat in the Supplementary Materials. These show that, in some cases, the path to the peak rate can be bumpy and the peak can be higher than the terminal response rate. This is evident in the parsed response rates of Rats 1–6 that were all trained with the lowest level of informativeness ($\iota=1.5$). By contrast, when informativeness is very high, the peak is usually reached almost immediately and there is little subsequent variation in the poke rate (e.g. see parsed rates for Rats 171–176 that were trained with the highest level of informativeness, $\iota=300$). The parsing of response rates shown in these figures used a stringent decision criterion; the algorithm found a step up or down only when the evidence for a divergence exceeded six nats. On the null hypothesis, a divergence that large would occur by chance about once in 1800 tests, well above the number of tests made when parsing the data over the 420 or 126 trials of the experiment. When the divergence between the contextual rate of reinforcement and the CS rate is high, as when $\iota=300$, there are very few steps (between 0 and 2) in the CS poke rate over 126 trials. By contrast, when the divergence between the contextual rate and the CS rate of reinforcement is low, as when $\iota=1.5$, there are many up and down steps, some lasting only a few trials. Thus, the linear increases to a peak revealed by the preceding analyses should not be taken to indicate that a steady increase is routinely seen in the individual subjects, regardless of the informativeness. When informativeness is low, the post-acquisition rate of responding during CSs and the difference between it and the rate during ITIs are unstable but trend upwards. When informativeness is high, a stable asymptotic rate generally appears after only one or two early steps, the first of which is often the step after the first trial.

One of us has previously argued that responding appears abruptly when the accumulated evidence that the CS reinforcement rate is greater than the contextual rate exceeds a decision threshold (*Gallistel et al., 2004*). The new, more extensive data require a more nuanced view. Evidence about the manner in which responding changes over the course of training is to some extent dependent on the analytic method used to track those changes. One method we have used here suggests that responding rises steadily over trials. The other method we have used relies on an information-theoretic measure of divergence to identify discrete points of change (up or down) in the response record. This method suggests the first increment in responding can be large and is usually followed by further, often smaller, increments in responding. At the same time, there is marked within-subject variability in the response rate, characterized by large steps up and down in the parsed response rates following the initial increment, but this variability tends to decrease across further training, with fewer and smaller steps in both the ITI and CS response rates. We think that the initial large increment reflects an underlying decision process whose latency is controlled by diminishing uncertainty about the two reinforcement rates and hence about their ratio. It is possible that diminishing uncertainty can also

explain the subsequent increments in responding (stepwise or gradual) as conditioning continues. We think that decreasing uncertainty about the true values of the estimated rates of reinforcement is also likely to be an important part of the explanation for the decreasing within-subject variability in response rates.

## Discussion

The present results have provided several important pieces of evidence about the nature of the conditioned response acquired by rats in an appetitive Pavlovian paradigm. The first is that the rate of learning, defined as the reciprocal of the number of trials needed to acquire a response to the CS, was determined by the CS's informativeness (equal to the $C/T$ ratio). Neither the rate at which the CS itself was reinforced, $1/T$, nor the spacing of the trials, $C$, had any independent effect after partialling out the effect of $C/T$. This is the same conclusion reached by *Gibbon and Balsam, 1981* in their meta-analysis showing scalar invariance in the acquisition of autoshaped key-pecking by pigeons. Contrary to what Gibbon and Balsam supposed, but in agreement with what *Jenkins et al., 1981* showed, the dependence of trials to acquisition on $C/T$ extends all the way to the abscissa, that is, when informativeness is very high acquisition can occur after just 1 reinforcement.

The second finding is that, when each CS is reinforced so that reinforcement rate scales inversely with $T$, the response rate after extended conditioning is directly proportional to $1/T$ (but not $C$ or $C/T$). More specifically, the CS response rate is a scalar function of the CS reinforcement rate (red symbols in *Figure 3*). Related to this is the third, and unexpected, finding that the response rate during the ITI is proportional to the contextual rate of reinforcement, $1/C$ (black symbols in *Figure 3*). It is particularly noteworthy that the same constant of proportionality relates the ITI response rates to the context reinforcement rate as relates the CS response rates to the CS reinforcement rates if the contextual rate of reinforcement is $0.5 \times 1/C$.

The fourth finding is that the post-acquisition response rate trends linearly upward to a peak. The higher the rate of reinforcement of the CS, the steeper that linear increase. However, the path to the peak rate has several ups and downs, particularly when informativeness is low, and the terminal rate can be lower than the peak rate. When informativeness is high, the response rate generally rises to the peak in one or two steps. The first and often final step often occurs after the first trial.

The previously published evidence concerning the relationship between $C/T$ and acquisition of Pavlovian conditioning with rodents has been mixed. *Lattal, 1999* and *Holland, 2000* reported evidence that conditioning in rats was related to $C/T$, as did *Ward et al., 2012* in a series of experiments with mice. However, *Kirkpatrick and Church, 2000*, and, more recently, *Thrailkill et al., 2020*, found no evidence for an effect of $C/T$ on trials to acquisition in rats. Moreover, Lattal, Holland, and Thrailkill et al. all found evidence for an effect of $T$ when $C/T$ was held constant. We suggest that these inconsistencies relate to differences in how response acquisition was measured across the studies. Our results show that the point at which evidence for conditioned responding first emerges is directly related to $C/T$, and not to $C$ or $T$ alone, but the level of responding subsequently acquired is directly related to $T$ and not to $C/T$ or $C$. Therefore, both $C/T$ and $T$ will affect any index of conditioning that is sensitive to both the time when responding emerges and how much responding is subsequently acquired. Lattal's evidence for an effect of $T$ between groups matched on $C/T$ was obtained in a test session conducted after 4 conditioning sessions totalling 48 trials, and Holland's evidence came from the last 8 of 16 conditioning sessions. In both cases, the observed effects of $T$ could have been due to its effect on the level of responding acquired rather than how quickly responding emerged. A similar argument can be made for the evidence provided by Thrailkill et al. based on the evidence that our rats took substantially longer to reach the Thrailkill et al. criterion for response acquisition than to reach the criterion we developed (see Supplementary materials, pages 7–8). This suggests that, to satisfy their criterion for acquisition, the rats acquired a higher level of responding which would have been affected by $T$. Finally, the absence of evidence for an effect of $C/T$ in the study by Kirkpatrick and Church may also have been due to the particular method they used to assess trials to acquisition. Their method, when applied to the present data (see Supplementary materials, pages 8–9), produced smaller differences between groups and the correlation between trials to acquisition and $\log(C/T)$ was not significant. Thus, their method may have been less sensitive to differences between groups in their rate of acquisition, which could explain why they failed to see an effect of $C/T$ across the relatively limited range of $C/T$ ratios they tested (from 1.5 to 12).

The results shown in **Figure 3** confirm that rats learn about rates of reinforcement and their response rate is directly proportional to the rate of reinforcement. This is very clear for responding during the CS, where the response rate is 22 times the rate of reinforcement in the CS (1 /T) across a 10-fold variation in reinforcement rate. Strikingly, the response rate in the ITI scales is by the same proportion to the overall reinforcement rate divided by 2 (i.e. ½ × 1 /C). This shows that rats compute the overall rate of reinforcement and, more specifically, their expectation of reinforcement in the context (when the CS is absent) is equal to approximately half the overall reinforcement rate. It has long been recognized that the contextual rate of reinforcement in appetitive operant conditioning has a scalar effect on foraging activities (**Belke, 1992**; **Drew et al., 2005**; **Killeen, 2023**; **Killeen et al., 1999**; **Killeen et al., 1978**). The effect of the operant contingencies is to channel that activity into the activity or activities on which reinforcement is contingent (**Gallistel and Shahan, 2024**). It now appears that the same is true in Pavlovian conditioning. Reinforcement is contingent on poking into the magazine, and the rats' poke rate is 22 times the expected reinforcement rate.

The contextual rate of reinforcement plays a fundamental role in Rate Estimation Theory (RET, **Gallistel and Gibbon, 2000**). It is the first term in the vector of *uncorrected* rates. The uncorrected vector is the list of the *observed* rates of reinforcement for each possible predictor (in our case, the CS and context). The *corrected* rate vector is the list of reinforcement rates subjects *ascribe* to the actions of the different predictors. The matrix equation that does the ascription is based on the assumption that they act independently, in which case the ascribed rates must sum to the observed rates when the predictors co-occur. The contextual rate of reinforcement also plays a fundamental role in the information-theoretic model of acquisition (**Balsam et al., 2006**; **Balsam and Gallistel, 2009**; **Ward et al., 2012**). In that model, the appearance of differential responding to the CS is determined by the informativeness of the protocol, which is the ratio between the CS reinforcement rate and the contextual reinforcement rate. The information transmitted to a subject by CS onset is the log of the informativeness.

## Information-theoretic contingency

Contingency in Pavlovian and operant conditioning has long resisted a mathematical definition that made it measurable in all circumstances, particularly when there is no time at which reinforcement may be anticipated, hence no time at which failures of reinforcement to occur can be counted (**Donahaoe, 2006**; **Gallistel, 2021**; **Gibbon et al., 1974**; **Granger and Schlimmer, 1986**; **Hallam et al., 1992**; **Hammond and Paynter, 1983**). **Gallistel and Latham, 2023** have developed a generally applicable measure based on the trivially computable prospective and retrospective mutual information between CSs and reinforcements or between responses and reinforcements. Given two distinguishable event streams X and Y—for example, a stream of CS onsets and a stream of reinforcements at CS termination—there is prospective mutual information between the x events and the y events, $I\left(\overrightarrow{X};Y\right)$, when the expected wait to the next y, conditional on an x, $(\mu_y|\overrightarrow{x})$, is reliably shorter than the expected wait between the y's $(\mu_y)$, as defined in **Equation 1**. There is retrospective mutual information, $I\left(X;\overleftarrow{Y}\right)$, when the expected wait looking back from a y to the most recent x, $(\mu_x|\overleftarrow{y})$, is shorter than the expected wait between the x's $(\mu_x)$, as in **Equation 2**:

$$I\left(\overrightarrow{X};Y\right) = \log\frac{\mu_y}{\mu_y|\overrightarrow{x}} = \log\frac{\lambda_y|\overrightarrow{x}}{\lambda_y} \tag{1}$$

$$I\left(X;\overleftarrow{Y}\right) = \log\frac{\mu_x}{\mu_x|\overleftarrow{y}} = \log\frac{\lambda_x|\overleftarrow{y}}{\lambda_x} \tag{2}$$

The arguments of the log function in **Equations 1; 2** are the ratio of the *unconditional wait* (in the numerator) to the *conditional wait* (in the denominator), or equivalently, the ratio of the inverse of the waits, *conditional rate* in numerator and *unconditional rate* in denominator. Because its logarithm is the mutual information, **Balsam et al., 2006** have termed this ratio the *informativeness* of a Pavlovian protocol.

**Gallistel and Latham, 2023** define *contingency* as the ratio of the *mutual information* to the *available information*, which is the amount that reduces subjective uncertainty to 0. Because measurement

error scales with latency (Weber's Law, see *Gibbon, 1977a*), contingency equals one only when the x's and y's coincide. How to measure the available information is often unclear. Mutual information, however, is trivially computed, as in *Equations 1; 2*.

Another information-theoretic measure, $n\mathrm{D_{KL}}$, measures the degree to which computed mutual information may be trusted. Its role, relative to mutual information, is analogous to the role of *p*-values relative to correlations. The $\mathrm{D_{KL}}$ in the $n\mathrm{D_{KL}}$ is the Kullback-Leibler *divergence*. It is analogous to the effect size in conventional statistics. The effect size is the normalized *distance* between two distributions assumed to have the same variance. Distance is symmetric, i.e., D(X,Y)=D(Y,X), but divergence is not: $\mathrm{D_{KL}}\left(X\|Y\right) \neq \mathrm{D_{KL}}\left(Y\|X\right)$. The asymmetric information-theoretic divergence is arguably the better measure, because the amount of data required to determine whether an X distribution differs from a Y distribution is not the same as the amount required to determine whether the opposite is true—and rats and mice are sensitive to this asymmetry (*Kheifets et al., 2017*; *Kheifets and Gallistel, 2012*).

The $\mathrm{D_{KL}}$ of one exponential distribution from another depends only on their rate parameters:

$$\mathrm{D_{KL}}\left(X\|Y\right)_{\exp} = \ln\frac{\lambda_X}{\lambda_Y} + \frac{\lambda_Y}{\lambda_X} - 1 \qquad (3)$$

The uncertainty regarding the values of estimates for $\lambda_X$ and $\lambda_Y$ depends on the sample sizes, $n_X$ and $n_Y$. Peter Latham (*Gallistel and Latham, 2023*, see their Appendix) has recently shown that when $\lambda_X = \lambda_Y$ (the null hypothesis),

$$n_e\mathrm{D_{KL}}\left(X\|Y\right) \sim \Gamma\left(n_p/2, 1\right), \qquad (4)$$

where $n_e$ is the *effective* sample size; $n_e = n_X/\left(1 + n_X/n_Y\right)$; and $n_p$ is the size of the parameter vector. For the exponential, $n_p = 1$. Therefore, as described earlier in *Equation 2*,

$$n\mathrm{D_{KL}}\left(X\|Y\right)_{\exp} = \frac{n_X}{1 + n_X/n_Y}\left(\ln\frac{\lambda_X}{\lambda_Y} + \frac{\lambda_Y}{\lambda_X} - 1\right) \qquad (5)$$

*Equation 5* measures how 'significant' an observed amount of mutual information is, how unlikely it is to have arisen by chance, and, by *Equation 4*, $n\mathrm{D_{KL}}\left(X\|Y\right)_{\exp} \sim \Gamma\left(0.5, 1\right)$, $n\mathrm{D_{KL}}$'s, so it may be converted to the more familiar *p*-values. Both are measures of the strength of statistical evidence. Conversion is motivated only by sociological considerations.

*Equations 1; 2; 5* and the RET equation—$\lambda_R = \mathbf{T}^{-1}\lambda_R^{\mathsf{T}}$ (*Gallistel, 1990*)—constitute a computationally simple, parameter-free model of associative learning. *Equation 2* enables us to address the question, How sensitive are subjects to the strength of the evidence for observed mutual information? Put another way, does their behavioural sensitivity to the accumulating mutual information suggest that they make a rational assessment of the extent to which they can trust the mutual information so far observed? In the next section, we propose a simple generative model of learning that incorporates answers to these questions.

## Towards a generative model that explains the learning rate law

Our results confirm a theoretically important law: The median learning rate in a group, operationally defined as the reciprocal of the median number of reinforcements prior to the appearance of the conditioned response, is an approximately scalar function of the protocol's informativeness, $\iota$, that is, of the ratio between the rate of reinforcement expected during the CS and the rate expected in the context in which the CS is presented. This law is descriptive; it does not model a hypothesised process that generates the behavioural data. As such, the law stands in contrast to formal models of the associative process, such as the well-known *Rescorla and Wagner, 1972* and its many descendants, that do propose a generative mechanism of the learning process. However, any generative model must explain the law described here. That necessity poses a seemingly insurmountable challenge to models in which reinforcements strengthen an associative bond (or augment a prediction) and non-reinforcements weaken the bond (or the prediction). The challenge arises from the fact that, when informativeness is substantial (≥5), a substantial fraction of subjects (23% or more) begin to respond appropriately to the CS after a single reinforcement (*Figure 2D*; see also *Appendix 1—figure 2D*). By contrast, when informativeness is low (≤3), no subject begins to respond after only one reinforcement, and almost all subjects take at least 10 reinforcements to begin responding.

After a single reinforcement, both the CS and the context have been reinforced once. Delta rule, or prediction-error, models for the updating of a hypothesised associative bond or prediction are event driven: the bond/prediction is augmented when a reinforcement occurs and decremented when an expected reinforcement fails to occur. The challenge is to explain why, when informativeness is high, the association/prediction between CS and reinforcement is already strong enough to be acted upon by many subjects after the first reinforcement. Decomposing the interval prior to the first CS into hypothesized 'trials' that weaken the context's association with reinforcement (cf. *Rescorla and Wagner, 1972*, p. 88) cannot offer an explanation because there can be no expectation of reinforcement prior to its first occurrence; failures of reinforcement cannot be imagined to have occurred during imagined trials prior to the first reinforcement. By contrast, when informativeness is low, a strong majority of subjects will not have shown evidence of selective responding to the CS even when the first 10 reinforcements have all occurred during the CS. A generative model must explain both extremes of the learning rate law.

A model of the behaviour-generating process must also generate the distributions of reinforcements to acquisition, not simply their central tendencies. Looking at *Figure 2*, we can see that, as informativeness increases, more and more data fall at the analytic lower limit of one reinforcement. Thus, the 'central' tendency is not in fact central; in the limit as informativeness becomes large, all the data fall at or above it. Statistically speaking, the distributions of reinforcements to acquisition become strongly left-skewed, with a mode at 1, the smallest possible value. The increasing left skew as $\iota$ increases in the distributions shown in *Figure 2D* is ascribable first to the analytic fact that statistically significant behavioural evidence that an association has been perceived can only exist when the number of reinforcements is $\geq 1$. It is ascribable second to the empirical fact that the ratio between the CS response rate and the contextual rate goes to 1 as the $\iota$ goes to 1. That is, the rate at which the behavioural evidence accumulates goes to 0 as the CS and contextual rates of reinforcement approach parity, which is the truly random control condition (*Rescorla, 1967*).

The distributions plotted in *Figure 2D* suggest to us a generative model incorporating an information-theoretic learning rule and performance function. In such a model, subjects perceive an association between the CS and reinforcement as soon as there is non-trivial evidence for it. The strength of that association is measured by the CS's informativeness, $\iota_{CS}$. A stochastic performance function then maps the strength of this perceived association into observable behaviour. The performance function can delay the appearance of measurable behavioural change, with the length of the delay decreasing as a monotonic function of the strength of the perceived association.

In the simplest model that predicts scaling between learning rate and informativeness, the probability that a subject changes its rate of responding after CS onset, $p(\Delta\lambda_r|CS\uparrow)$, as a function of the CS's informativeness, is described by the exponential cumulative distribution function

$$p\left(\Delta\lambda_r|CS\uparrow\right) = 1 - e^{-k\cdot(\iota_{CS}-1)}. \tag{6}$$

This function describes how the probability of a response increases as $\iota_{CS}$ increases, and thus the median number of trials to see a response decreases with increasing $\iota_{CS}$. As the contingency between CS and reinforcement goes to zero and $\iota_{CS}-1$ goes to zero, $p(\Delta\lambda_r|CS\uparrow)$ goes to 0. In *Figure 5A*, the cumulative probability of seeing at least one response across trials is plotted for each of the 14 levels of informativeness in the current experiment, using $k=1/297$ from the regression equation in *Figure 1*. The horizontal distance between the curves corresponds to the effect of informativeness on the number of trials required to reach a response criterion. For example, $p(\Delta\lambda_r|CS\uparrow)=0.5$ after 1 trial when $\iota=300$ but after about 100 trials when $\iota=3$. These values, corresponding to the number of trials for the median subject to make at least one response, are plotted as a black line in *Figure 5B*. This is equivalent to the regression in *Figure 1* with slope $-1$ in the log-log domain. In this model, the appearance of conditioned responding is delayed when $\iota$ is low because the probability of responding on any trial is low.

In the remaining panels in *Figure 5* (C to I), the modelled probability of a response at different levels of informativeness is compared with the data from *Figure 2* showing the proportion of rats that had acquired responding to the CS at the same informativeness, based on the criterion that the CS response rate was permanently greater than the overall response rate. In most cases, the modelled probabilities of a response correspond well with the observed proportion of rats that had acquired a response. Thus, the simple generative model of responding in *Equation 6*, with only one parameter

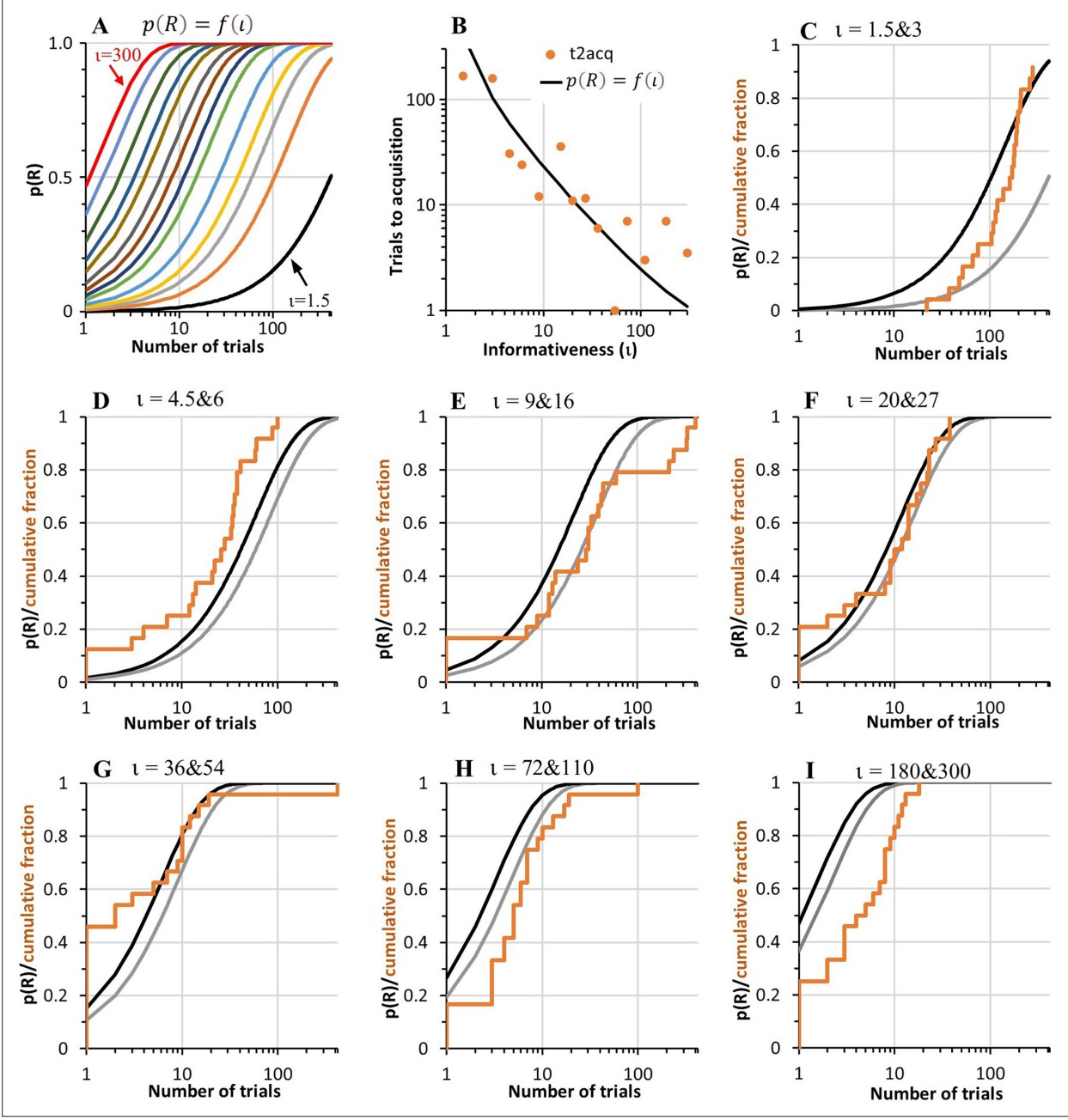

**Figure 5.** Modelling response probability across trials as a function of informativeness. Note A: The cumulative probability of a response, p(R), calculated using **Equation 6**, as a function of number of reinforced trials. Each curve plots p(R) for one of the 14 levels of informativeness used in the current experiment, ι =1.5–300.B: The black line plots the number of trials taken for p(R) to reach 0.5 as a function of the same 14 levels of informativeness. The plot includes the median trials to acquisition (t2acq) for the 14 groups in the current experiment, replotted from **Figure 2A**. DtoI: The cumulative fraction of rats that had acquired responding (orange lines) for pairs of groups with similar informativeness, replotted from **Figure 2D**. The black and grey curves plot the cumulative p(R) using the same two values of ι (the black line is based on the higher of the two values).

that we have fixed at k=1/297 based on data reported previously by *Gibbon and Balsam, 1981*, can explain with remarkable success the learning rates of the 14 groups in our experiment, predicting not only the central tendencies of the groups but also their distributions.

## Conclusion

Associative learning obeys simple equations that map from measurable properties of subjects' experience to measurable properties of their behaviour. The equations have at most one free parameter, and

it is a scale factor with a data-anchored interpretation. According to one of these rules, the response rate is 23 times the reinforcement rate in the median subject (**Figure 3**) over a range that covers three orders of magnitude. The variability about this regression is also multiplicative, with a scale factor of $10^{.39}$=2.5. A second rule is the simple scalar relation between the learning rate (the reciprocal of reinforcements to acquisition) and the informativeness of a Pavlovian protocol. In its simplest form, the log of the learning rate is –1 times the mutual information; the constant, *k*, in this regression is the informativeness that produces 1-trial acquisition (**Figure 1** and **Figure 2**). A third example is the parameter-free rate estimation equation. In a 1-CS protocol, that equation is:

$$\begin{bmatrix} \hat{\lambda}_{R|C\&\sim CS} \\ \hat{\lambda}_{R|CS} \end{bmatrix} = \begin{bmatrix} 1 & 1/\iota \\ 1 & 1 \end{bmatrix}^{-1} \begin{bmatrix} \lambda_{R|C} \\ \lambda_{R|C\&CS} \end{bmatrix} \tag{7}$$

*Equation 7* maps the *observed* contextual rate of reinforcement, $\lambda_{R|C}$, and the *observed* rate of reinforcement when the CS is also present, $\lambda_{R|C\&\sim CS}$, to the rates of reinforcements that subjects *ascribe* to these predictors, $\hat{\lambda}_{R|C\&\sim CS}$ and $\hat{\lambda}_{R|CS}$. It does so by way of the inverse of a simple matrix. The only element not equal to 1 in the matrix in *Equation 7* is the inverse of the informativeness, which is also the protocol parameter in the learning-rate equation. *Equation 7* explains the results in the cue-competition literature (a.k.a. the assignment-of-credit literature). (For review and comparison to the Rescorla-Wagner model, see **Gallistel, 1990**, Chapter 13).

These equations bring out the formal structure of the *data* from associative learning experiments. They map an always defined and computable aspect of subjects' experience—*rate* of reinforcement— to measurable properties of subjects' behaviour. They contrast with equations that map from the often-undefinable *probability* of reinforcement to a *hypothetical construct* like associative strength or expected value (**Honey et al., 2020**; **Ludvig et al., 2012**; **Pearce and Hall, 1980**; **Rescorla and Wagner, 1972**; **Sutton and Barto, 1990**; **Vogel et al., 2019**).

That the rates of reinforcement rather than the probabilities are the inputs to the equations that map perceived associations to behaviour calls into question the basic assumption in the common under-standing of associative processes, the assumption that associative learning is mediated by the incre-menting and decrementing of some scalar brain quantity (e.g. the conductivity of a plastic synapse) by reinforcement and non-reinforcement, respectively. This assumption is central to all models of Pavlovian and operant learning that map *probability* of reinforcement to associative strength and is central to all reinforcement learning models that map it to a weight state in a neural net (supervised and unsupervised learning) or to an expected long-term reward (reinforcement learning). They all fail to 'leverage the statistical and computational structure of [the] problem' (**Russo et al., 2018**, p. 6), because they must discretize time into unobserved trials or states with unspecified durations (**Namboodiri, 2022**). They must do this because, unlike a rate that has a temporal unit, a probability is unitless.

A probability is the ratio between a count of reinforcements and the sum of the count of reinforce-ments and non-reinforcements. It is impossible to map from probability of reinforcement to behaviour in the general case because the non-reinforcements in the denominator are unobserved events with no physical properties and, in the general case, often uncountable. They are countable only when reinforcement fails to occur at an expected time of reinforcement. In our protocol and many others, the reinforcements occur at an unpredictable time or times following CS onset. The durations of our CS were drawn from a uniform distribution, and the reinforcements coincided with its termination. In **Rescorla, 1967**; **Rescorla, 1968** experiments with truly random controls (and the many follow-ons), the reinforcements were programmed by Poisson processes. The defining feature of a Poisson process is its flat hazard function: there is no moment at which a reinforcement is any more likely than at any other moment.

Unlike models that take probability of reinforcement as the essential aspect of subjects' experi-ence, all of which have at least two free parameters, the model of associative learning provided by the simple equations we present here directly explains quantitative facts about associative learning that have gone unexplained for decades: One such fact is that reinforcements to acquisition are unaf-fected by partial reinforcement (**Balsam and Gallistel, 2009**; **Gallistel, 2003**; **Gallistel et al., 2014**; **Gibbon et al., 1980**; **Gottlieb, 2004**; **Gottlieb, 2005**). This follows immediately from the equation

for the learning rate as a function of informativeness (top of *Figure 1*), because partial reinforcement does not alter informativeness. A second such fact is that deleting reinforced trials while retaining the spacing of the remaining reinforced trials does not alter the temporal progress of acquisition (*Bouton and Sunsay, 2003*; *Gottlieb, 2008*). The number of reinforced trials in a given amount of training time is irrelevant, given only that there is at least one. This, too, follows from the equation that relates the learning rate to the informativeness or to its log, the mutual information (*Figure 1* and *Figure 2*). Because the slope on a log-log plot is essentially –1 over most of the usable range of values for informativeness, halving the number of trials doubles the informativeness, and that doubles the learning rate.

We conclude that models of associative learning based on probability of reinforcement cannot be correct because the rate of responding, the learning rate, and the assignment of credit are simple functions of rate of reinforcement, a quantity with temporal units. Only models that leverage the metric temporal structure of subject's experience can be neurobiologically realisable.

## Methods
### Subjects
A total of 176 experimentally naive female albino Sprague Dawley rats (8–10 weeks of age) were obtained from the Animal Resources Centre, Perth, Western Australia. They were housed in groups of 4 in split-level ventilated plastic tubs (Techniplast), measuring 40×46 ×40 cm (length × width × height), located in an animal research facility at the University of Sydney. They had unrestricted access to water in their home tubs. Three days before commencing the experiment, they were placed on a restricted food schedule. Each day, half an hour after the end of the daily training session, each tub of rats received a ration of their regular dry chow (3.4 kcal/g) equal to 5% of the total weight of all rats in the tub. This amount is approximately equal to their required daily energy intake (*Rogers, 1979*) and took at least 2 hr to be eaten (but was usually finished within 3 hr). Rats on this schedule do not typically lose weight (and never more than 10%) but gain weight only very slowly. All experimental procedures were approved by the Animal Research Authority of the University of Sydney (protocol 2020/1840).

### Apparatus
Rats were trained and tested in 32 Med Associates conditioning chambers distributed equally across four rooms. Twenty-four chambers (Set A) measured 28.5×30 x 25 cm (height × length × depth) and the other eight (Set B) were 21×30.5 ×24 cm (height × length × depth). Each chamber was individually enclosed in a sound- and light-resistant wooden shell (Set A) or PVC shell (Set B). The end walls of each chamber were made of aluminum; the sidewalls and ceiling were Plexiglas. The floor consisted of stainless-steel rods, 0.5 cm in diameter, spaced 1.5 cm apart. Each chamber had a recessed food magazine in the center of one end wall, with an infrared LED and sensor located just inside the magazine to record entries by the rat. A small metal cup measuring 3.5 cm in diameter and 0.5 cm deep was fixed on the floor of each food magazine either in the center (Set A) or offset to the left of center (Set B). Attached to the food magazine was a dispenser delivering 45 mg food pellets (purified rodent pellets; Bioserve, Frenchtown, NJ). Illumination of an LED (Med Associates product ENV-200RL-LED) mounted in the ceiling of the magazine served as the CS. Experimental events were controlled and recorded automatically by computers and relays located in the same room. Throughout all sessions, fans located in the rear wall of the outer shell provided ventilation and created background noise (between 61 and 66 dB, depending on the chamber).

### Procedure
Each tub of four rats was randomly allocated to one of the 14 groups shown in *Table 1*. The experiment was run with three separate cohorts of 64, 80, and 32 rats, so that by the end of the experiment, each of the 14 groups had at least 12 rats (Groups 6 and 10 had 16 rats, as described next). In the first cohort, the data for 4 rats in each of Groups 6 and 10 were not saved in Session 1 due to human error. These eight rats continued through the entire experiment, but only their data for the final five sessions were used (to calculate their terminal response rate). An extra four rats were run in both groups in the second cohort.

The rats were not given magazine training. The experiment commenced with the first conditioning session and continued for 42 sessions over 42 consecutive days. Within each group, the CS-US interval varied from trial-to-trial according to a uniform distribution centred on the value of $T$ for that group (the interval varied from a minimum of 2 s to a maximum of 2x$T$–2 s). The ITI in each group also varied from trial-to-trial as a uniform distribution with a minimum of 15 s. Between groups, $T$ varied from 6 s to 62 s, and $C$ varied from 63 s to 4200 s (70 min) as summarised in *Table 1*. The combinations of $C$ and $T$ gave rise to 14 distinct $C/T$ ratios that were approximately evenly distributed on a log scale ranging from a ratio equal to 1.5 (63/42) up to 300 (4200/14). For the first 11 groups (those with ratios from 1.5 to 72), there were 10 trials per session; for the remaining three groups (with $C/T$ ratios of 110, 180, and 300), each session contained only three trials so that the total session time remained within a manageable length (less than 4 hr). All groups were trained with one session per day for a total of 42 sessions. Photo-beam interruptions by entry into the magazine were recorded during each CS and each ITI (recorded during the 10 s period immediately before CS onset).

## Data analysis

Several different indices were used to identify when responding to the CS first appeared. The first index involved creating, for each rat, cumulative records of response counts during the CS and overall response rates (across the CS and ITI), then converting these into cumulative records of response rates by dividing the cumulative count at each trial by the cumulative time (CS or CS + ITI) at that trial. Since the response rate during the CS should not differ from the overall rate before the rat has learned that the CS signals food, we identified the point of acquisition as the trial, $t_i$, for which the cumulative response rate to the CS after that trial (starting from $t_i$ +1) was permanently greater than the cumulative overall response rate starting from $t_i$ +1. This trial is plotted for each rat in *Figure 2A*. Next, we used a new information-theoretic statistic, the $n\mathrm{D_{KL}}$ (see next section), to identify when, starting from trial $t_i$ +1, the cumulative rate of responding during the CS began to exceed the overall rate by a statistical threshold. *Figure 2B* shows the trials to acquisition when the computed $n\mathrm{D_{KL}}$ exceeded a threshold of 0.82, which corresponds to odds of 4:1 in favour of a difference between the CS and overall response rates. *Figure 2C* shows the trials to acquisition when the $n\mathrm{D_{KL}}$ exceeded 1.92, corresponding to probability $p<0.05$ that the CS and overall response rates are the same. Additional analyses were run adopting measures used in previous studies (*Kirkpatrick and Church, 2000*; *Thrailkill et al., 2020*) to identify trials to a learning criterion. These analyses and results are described in the Supplementary Materials.

Analyses were also conducted to assess how $T$, $C$, and $C/T$ affect conditioned responding after the point when it first emerges. These analyses examined how the response rate increased over trials and what level of responding was reached after extended training. The level of responding ultimately acquired by each group was computed as the mean response rate over the last 5 conditioning sessions. Further analyses (shown in *Appendix 1—figure 2*) broke down the response rate into separate components: the latency to first response; the mean duration of each response (time in the magazine); and the interval between responses (time out of the magazine). The increase in responding over sessions was assessed using a method described by *Harris, 2022* that uses the slope of the cumulative response record to identify the number of trials required for responding to reach successive response milestones corresponding to each decile of the rat's peak response rate. Based on findings relating response rates to reinforcement rates (1 /$T$) using this paradigm (*Harris and Carpenter, 2011*), we hypothesised that responding at the end of conditioning would be related to $T$, and indeed would scale linearly with log($T$) but would not be related to $C$ or $C/T$. We had no clear hypotheses about whether $T$, $C$, or $C/T$ would affect how quickly responding increases after it has emerged.

## The Kullback-Leibler divergence and the $n\mathrm{D_{KL}}$

Comparing the CS poke rate to the ITI poke rate reinforcement-by-reinforcement is problematic in cases where one or both rates are undefined because the subject has not yet made a poke. When a subject has made five pokes during the first 2 CSs and no pokes during the much longer ITIs, one does not want to conclude that the subject has not yet acquired a conditioned response to the CS. A second problem when using conventional statistics like the $t$-test is that one is required to specify in advance the sample size.

We circumvented these difficulties by reformulating the null hypothesis as a comparison between the rate of responding during the CS and the overall contextual rate of responding across the whole trial (during the ITI and CS) by using an information-theoretic statistic to measure the strength of the evidence that the CS rate of responding differs from the contextual rate (*Gallistel and Latham, 2023*). The information theoretic statistic measures the strength of the evidence against the hypothesis that the distribution of inter-poke intervals during CSs is the same as the distribution in the context overall. Put another way, the null hypothesis is that poking during CSs does not differ from the poking expected because the subject pokes into the magazine in response to the fact that pellets sometimes drop there without regard to the signal value of the CS.

The CS poke rate, $\lambda_{r|CS}$, at any point in training is the number of pokes made during the CSs divided by their cumulative duration: $\lambda_{r|CS} = n_{r|CS}/D_{CS}$. And likewise for the contextual poke rate: $\lambda_{r|C} = n_{r|C}/D_C$. In these calculations, $D_{CS}$ is the cumulative duration of the CS and $D_C$ is cumulative training time. Assume, for example, that cumulative CS time as of the second reinforcement is 20 s, total training time is 1000 s, and the subject has made 5 pokes during the two CS intervals and no pokes during the ITIs. Then, $\lambda_{r|CS} = 5/20 = .25/s$ and $\lambda_{r|C} = 5/1000 = .005/s$. The ratio of the two estimates is 0.25/0.005=50:1, that is, the subject's observed poke rate at this point in training is 50 times faster during the CSs than would be expected given the estimate of how frequently it pokes in the training context. This discrepancy is unlikely to have arisen by chance.

An information theoretic statistic, the $n\mathrm{D}_{KL}$, measures the unlikeliness (*Gallistel and Latham, 2023*). The $\mathrm{D}_{KL}$ is the Kullback-Leibler divergence, a measure of the distance between two distributions. The divergence of one exponential distribution from another depends only on the rate parameters of the distributions. Therefore, as per *Equation (3)*, the divergence of $\lambda_{r|CS}$ from $\lambda_{r|C}$ is

$$\mathrm{D}_{KL}\left(\lambda_{r|CS}\|\lambda_{r|C}\right)_{exp} = log_e\left(\frac{\lambda_{r|CS}}{\lambda_{r|C}}\right) + \frac{\lambda_{r|C}}{\lambda_{r|CS}} - 1$$

It is the information-theoretic measure of the extent to which an exponential distribution with rate parameter $\lambda_{r|CS}$ diverges from an exponential distribution with rate parameter $\lambda_{r|C}$. The divergence is, roughly speaking, the equivalent of the effect size in a conventional analysis. The equivalence is rough because the effect size—the normalized distance between the means—is symmetric, whereas the divergence is not: $\mathrm{D}_{KL}\left(\lambda_{r|CS}\|\lambda_{r|C}\right) \neq \mathrm{D}_{KL}\left(\lambda_{r|C}\|\lambda_{r|CS}\right)$.

The $n\mathrm{D}_{KL}$ measures the additional cost of encoding $n$ data drawn from the exponential distribution with parameter $\lambda_{r|CS}$ on the assumption that they come from an exponential distribution with rate parameter $\lambda_{r|C}$ (*Cover and Thomas, 1991*). It multiplies the $\mathrm{D}_{KL}$ by the effective sample size, $n = n_{r|CS}/\left(1 + n_{r|CS}/n_{r|C}\right)$. Therefore, as per *Equation 5*,

$$n\mathrm{D}_{KL}\left(\lambda_{r|CS}\|\lambda_{r|C}\right) = \frac{n_{r|CS}}{1 + n_{r|CS}/n_{r|C}}\left(log_e\left(\frac{\lambda_{r|CS}}{\lambda_{r|C}}\right) + \frac{\lambda_{r|C}}{\lambda_{r|CS}} - 1\right).$$

When there is no divergence, the $n\mathrm{D}_{KL}$ is distributed gamma(.5,1) (for proof, see Appendix in *Gallistel and Latham, 2023*). Thus, we can convert the information-theoretic measure of the strength of the evidence for divergence to the more familiar *p*-value measure.

In the illustrative example, $\mathrm{D}_{KL}\left(.25\|.005\right) = 2.93$ nats, and n=5/ (1+5/5)=2.5. (The nat is a unit of information equal to the base of the natural logarithm. The Kullback-Leibler divergence must be computed using the natural logarithm, but the result in nats may be converted to bits by multiplying by $log_2 e = 1.44$.) Therefore, the $n\mathrm{D}_{KL} = 2.5 \times 2.93 = 7.32$ nats and $1 - \mathrm{gamcdf}\left(7.32, .5, 1\right) \cong .0001$. The evidence against the null hypothesis is very strong; the odds against the assumption that all 5 pokes have occurred only during the CSs only by chance are on the order of 10,000 to 1. This example illustrates one approach to the statistical comparison of poke rates following each of the first few variable duration CSs, each terminating in a reinforcement.

## Parsing response rates

Another approach to estimating the onset of conditioned poking comes from parsing the CS and ITI inter-poke interval vectors into segments with significantly different rate parameters. Parsing of the inter-poke interval vectors was also motivated by a desire to capture differences between subjects in the often bumpy evolution of their post-acquisition rate of responding.

Our parsing algorithm recursively extends the length, $n_e$, of the vector of inter-poke intervals one interval at a time. After each extension, it compares the rate estimate for each successively longer sub-sequence to the rate estimate for the full sequence, using the $nD_{KL}$ statistic. These comparisons generate the function $nD_{KL}(n_s)$ for $n_s \leq n_e$. Whenever $\max\left[nD_{KL}(n_s)\right] > c$, the parser truncates the inter-poke interval vector at the location of the maximum. The value for the decision criterion, $c$, is user-supplied and generally falls between 2 and 6 nats. The rate estimate for the segment truncated is the number of pokes up to the truncation divided by the sum of the intervals up to the truncation. The algorithm operates recursively on the post-truncation portions of the vector until there is no significant sub-sequence. The results for values of 2, 4, and 6 nats, corresponding to $p$-values 0.05, 0.005, and 0.0005, are contained in an Excel file, ParseTable.xlsx, that can be downloaded from https://osf.io/vmwzr/ along with the MATLAB code for the custom expparser.m function. Because parsing inescapably involves multiple comparisons, highly conservative decision criteria are generally to be preferred. When reporting results, we give only the results for $c$=6 nats.

In computing the parses, we did not include the first CS poke in any given CS in our estimate of the CS poke rate because the latencies to the first pokes clearly came from a different distribution than the distribution of inter-poke intervals. In protocols with a short mean CS, the mean inter-poke interval (the reciprocal of the poke rate) was as short as 0.1 s, whereas the mean latency to the first poke in a CS was rarely shorter than 2 s. The explanation for the slow first pokes is probably the fact that the duration of a CS—hence the minimum reinforcement latency—was never shorter than 2 s. In the denominator of our poke rate estimates, we excluded the interval to the first poke. We also excluded the intervals when the head was in the hopper, because a poke cannot be made when the head is in the hopper. CS durations with no pokes were also excluded because there was no way to estimate the first-poke latency. Thus, our estimates of the poke rates included only CSs where there were pokes and the $n$'s in the numerators of those estimates were only the counts after each first poke. In the denominator was the cumulative duration of the CSs that had at least one poke minus the cumulative latencies to the first pokes minus the cumulative duration of the hopper entries. The estimates of the poke rates during the pre-periods were the cumulative number of pokes in those intervals divided by their cumulative duration. The estimates of the contextual poke rates were the Pre poke rate estimate and the CS poke rate estimate weighted, respectively, by the cumulative ITI duration divided by the cumulative training duration and by the cumulative CS duration divided by the cumulative training duration.

The initial rate estimate in a parse extends back to the beginning of observation, which makes it possible to plot, for example, an initial ITI poke rate that starts at the beginning of training even though the first poke during a Pre interval may not have occurred until the fifth ITI. The parsing gives an alternative way of comparing rates of poking as of each successive reinforcement for the first few reinforcements. This becomes important when informativeness is high because then conditioned poking appears very early in training, as early as the second CS presentation.

Given the novelty of the methods used to estimate the onset of conditioned poking, we thought it essential to provide plots of the results on which these estimates are based. The Supplementary Materials (pages 10–185) include, for every rat, a plot of the $nD_{KL}$ and cumulative response rates during the CS, ITI, and context, as well as plots of the parsed response rates during the CS and ITI. **Figure 6** provides examples of these plots for two rats: Rat 3, trained with the lowest level of informativeness ($\iota$ =1.5), and Rat 176, trained with the highest level of informativeness ($\iota$ =300). The top row of plots shows the cumulative response rates during the CS, $\lambda_{r|CS}$, during the ITI, $\lambda_{r|ITI}$, and the overall rate in the context, $\lambda_{r|C}$, as well as the signed $nD_{KL}$ for the divergence between CS and context response rates. The bottom row of plots shows the parsed response rates for the CS and ITI.

In the top two plots of data from Rat 3, all three response rates rise rapidly over the first 40 trials and then decline. The ITI rate (dashed line) is generally greater than the CS rate (solid line) until about the 200th trial in **Figure 6A** (or 110th trial when using the adjusted measure of response rate in 6B), after which it is consistently lower. Both averages steadily decline up to about Trial 300, implying a persistent drop in both rates following their peak at around Trial 40. At about Trial 300, the average CS poke rate begins to rise rapidly, implying that the CS poke rate increased at the start of the rise. The slight rise in the dashed curve indicates a slight increase in the pre-poke rate as well. The successive changes in the poke rates inferred from these plots of the cumulative rate estimates are confirmed by the parses plotted in **Figure 6C and D**, where the solid black line plots the parse of the CS poke rate

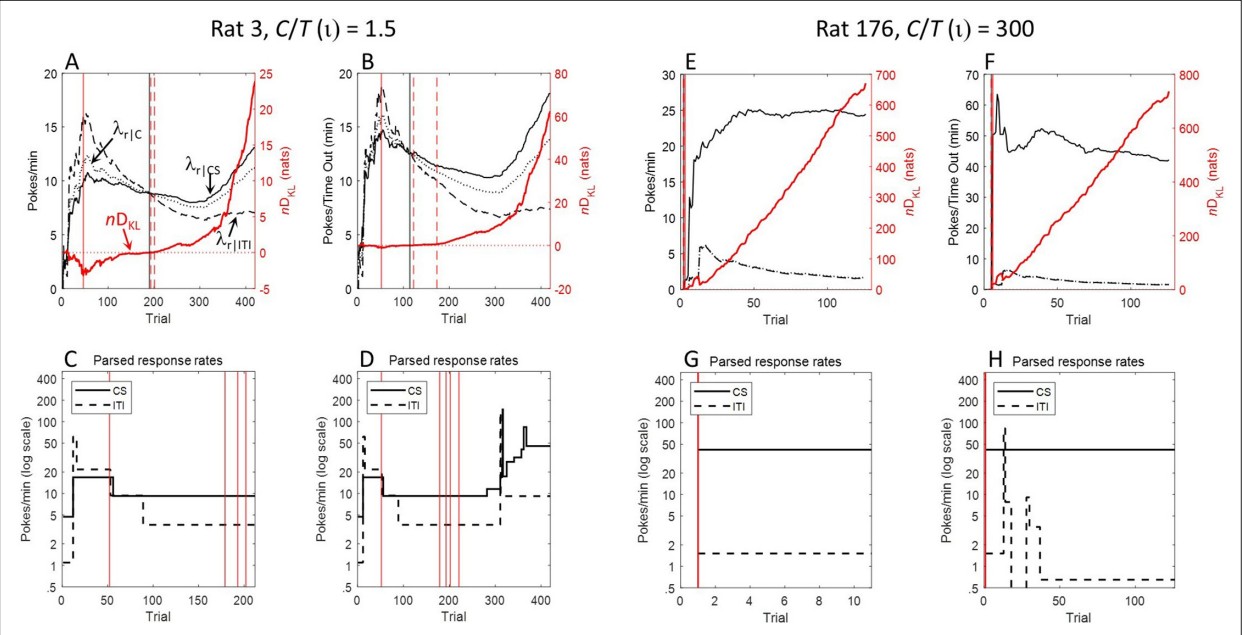

**Figure 6.** $nD_{KL}$ and parsed response rates for two rats. Note: Panels A to D show data from Rat 3; panels E to H show the corresponding data from Rat 176. For each rat, the top two panels plot the cumulative rate of poking during the conditioned stimulus (CS), $\lambda_{r|CS}$ (solid black line), during pre-CS inter-trial intervals (ITIs), $\lambda_{r|ITI}$ (dashed black line), and the contextual rate, $\lambda_{r|C}$ (dotted black line), as functions of the number of trials (1 reinforcement per trial). In each plot, the red curve is the signed $nD_{KL}$ plotted against the right axis. The black vertical line marks when the cumulative CS response rate permanently exceeded the cumulative context rate. The two red dashed vertical lines to the right of the black line mark when the $nD_{KL}$ reached 0.82 (Odds 4:1 that CS rate >Context rate) and 1.92 (p<0.05 that CS rate = Context rate), and the unbroken red line marks the minimum of the $nD_{KL}$. In A and E, response rates were calculated conventionally, as number of responses divided by total time. In B and F, response rates were calculated as the number of responses excluding the first response in each CS divided by the remaining time (after the first response) out of the magazine. The bottom two panels for each rat show the parsed estimates of $\lambda_{r|CS}$ and $\lambda_{r|ITI}$. In C and G, the x-axis has been right-cropped to better reveal early changes. In all plots, the vertical red lines mark estimates of acquisition based on the $nD_{KL}$ as the Earliest estimate (leftmost), and when the odds against the null hypothesis that the parsed CS and ITI rates were equal reached 4:1, 10:1, 20:1, and 100:1.

and the dashed black line the parse of the Pre rate. The pre-poke rate started lower than the CS rate but jumped after about 10 reinforcements to a higher value. At about Trial 40, both rates became the same, but at about Trial 85, the Pre-poke rate dropped permanently below the CS rate. At about Trial 300, there is a sequence of increases in the CS poke rate, accompanied by a single smaller increase in the Pre poke rate. The decision criterion used in this and all the plotted parses was 6 nats, so one may have substantial confidence that the changes are statistically significant. (When $nD_{KL}$ = 6, the odds are greater than 2000:1 against the null hypothesis.)

The signed $nD_{KL}$ function is plotted in red against the right axis in the top row of **Figure 6**. The $nD_{KL}$ is always positive because the magnitude of a divergence cannot be negative. The direction of a divergence may, however, vary; the Pre rate may be greater than or less than the CS rate. To make the direction of divergence apparent, we give the $nD_{KL}$ positive sign when the CS rate is greater than the Context rate and negative sign when the reverse is true. For Rat 3, there is a short initial positive spike in the signed $nD_{,KL}$ followed by an interval of several tens of reinforcements when it is negative. After about 40 reinforcements, it hits its minimum and began a more or less steady climb. Its last upward crossing of the 0 line is at about Trial 200 (Trial 110 for the plot to the right).

Rat 176, whose data are plotted in **Figure 6E–H**, made no pokes during the first 2 CSs, one during the third CS, none during the **fourth,** one during the fifth, and 16 on the 6th. It made no pokes during the first five 30 s pre-intervals. Consequently, the average ITI poke rates and the average contextual poke rates are undefined over the first few trials and so is the $nD_{KL}$. On Trial 3, the CS response rate and Context response rate can be defined—and the $nD_{KL}$ is already greater than 2 (thus p<0.05). By Trial 6, the $nD_{KL}$ is 11.7 nats, which corresponds to odds of 760 million to 1 against the null. Despite this, one might conclude there was no evidence of CS-conditional responding in this subject until Trial 3.

However, as already remarked, the first segment of a rate parse always extends back to the onset of observation. In the parses of the CS poke rate and the ITI poke rate, which are plotted in *Figure 6G and H*, there is no evidence for a change in the poke rates at Trial 3. Indeed, the parse of the CS poke rate shows no change over all 126 CSs, while the parse of the ITI poke rate finds the first change to be at Trial 13.

The failure to find changes in the poke rates at Trial 3 or even 6 is not a consequence of the high value for the decision criterion. Lowering it from 6 nats to 4 did not change the parse; lowering it to 2 nats produced a parse with a single change, a drop from 52 pokes/min to 38 pokes/min at Trial 40. Likewise, for the parse of the ITI rate of poking: lowering the decision criterion from 6 nats to 2 nats did not alter it. Nor does this failure occur because the algorithm cannot find very short segments. Parses of the contextual poke rate at all three values for the decision criterion find the same 1-trial long segment at Trial 13—see upward blip in dashed last plot in *Figure 6*. In other subjects, upward or downward blips 1 or 2 trials long are sometimes found at the outset of training.

There are significantly fewer pokes during the first 5 CSs than expected given the initial rate estimate for the CS poke rate. Much of this is attributable to the initially slow reaction to CS onset. The poke on Trial 3 came 20.3 s into the 23 s long CS; the poke on Trial 5 came 14.45 s into the 16 s long CS. The latency to make the first poke dropped rapidly over the first 9 trials from a mean greater than 11 s for the first 5 trials to a mean of 3.6 s for the Trials beyond 10. During the first five trials, the observation intervals during which it was possible to register a poke that would go into the parsing algorithm totaled only 4.18 s. Given an initial poke rate estimate of 0.7 /s, the expected number of pokes in that interval is 2.9, and there is a 5% chance of observing no pokes.

All considered, an argument can be made in this and several other cases where the informativeness was large that the parse results are a better indicator of the onset of CS-conditional poking, particularly when informativeness is high and the ITI poke rate is very low. Blips in the ITI poke rate, often put it momentarily greater than the CS poke rate (see dashed line in *Figure 6H*), are common in post-acquisition protocols. Therefore, our parse-based estimate of the onset of conditioning is the trial after which the parsed CS poke rate is greater than the parsed ITI poke rate on 95% of the trials.

## Acknowledgements

This research was supported by funding from the Australian Research Council, grant DP210102343. The data and analysis code presented in this article are available for download at https://osf.io/vmwzr/. The work was not pre-registered.

## Additional information

### Funding

| Funder | Grant reference number | Author |
| --- | --- | --- |
| Australian Research Council | DP210102343 | Justin A Harris |

The funders had no role in study design, data collection and interpretation, or the decision to submit the work for publication.

### Author contributions

Justin A Harris, Conceptualization, Resources, Data curation, Formal analysis, Funding acquisition, Investigation, Visualization, Methodology, Writing – original draft, Project administration, Writing – review and editing; Charles Randy Gallistel, Conceptualization, Data curation, Software, Formal analysis, Visualization, Writing – original draft, Writing – review and editing

### Author ORCIDs
Justin A Harris ⓘ https://orcid.org/0000-0003-3865-8097
Charles Randy Gallistel ⓘ https://orcid.org/0000-0002-4860-5637

### Ethics

All experimental procedures were approved by the Animal Research Authority of the University of Sydney (protocol 2020/1840).

Reviewer #1 (Public review): https://doi.org/10.7554/eLife.102155.4.sa1
Reviewer #2 (Public review): https://doi.org/10.7554/eLife.102155.4.sa2
Author response https://doi.org/10.7554/eLife.102155.4.sa3

---

## Additional files

### Supplementary files

MDAR checklist

Supplementary file 1. Data plots for each individual rat.

### Data availability

All data and analysis code are shared in https://osf.io/vmwzr/.

The following dataset was generated:

| Author(s) | Year | Dataset title | Dataset URL | Database and Identifier |
| --- | --- | --- | --- | --- |
| Harris J, Gallistel CR | 2025 | Information, certainty, and learning | https://osf.io/vmwzr/ | Open Science Framework, vmwzr |

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

## Appendix 1

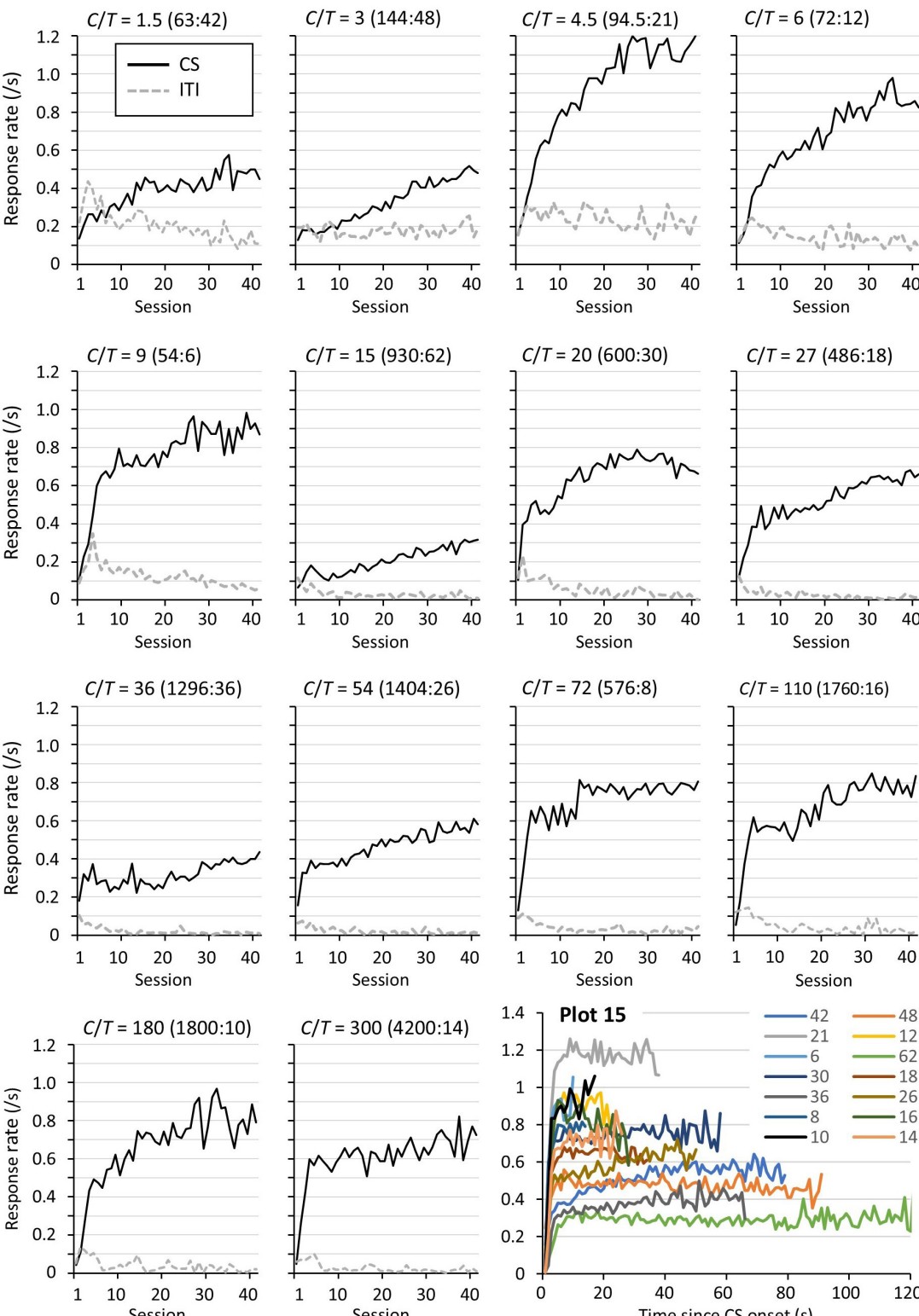

**Appendix 1—figure 1.** Mean response rate for each group across Sessions or across time since CS onset. There were 10 (groups with C/T ≤ 72) or 3 (groups with C/T ≥ 110) trials per session. Plot 15 (bottom right) shows the mean response rate per second during the CS, averaged from the last 5 sessions. Each group is identified by the length of the mean CS-US interval (T).

## Analyses of terminal response rates (last 5 sessions)

As a measure of the terminal level of responding, the response rates during the CS were averaged over the last 5 sessions for each rat. We have conducted the same analysis described here using the data from the final 10 session, with very similar results. (Response rate data of individual rats averaged from both the last 5 and last 10 sessions can be downloaded from https://osf.io/vmwzr/) We first calculated CS response rate as total number of responses during the CS (summed across all trials over the 5 sessions) divided by the total CS duration (summed across all trials over the 5 sessions). We then compared this terminal response rate with the log of $T$, $C$, and $C/T$ ($\alpha = 0.017$ after correction for multiple comparisons). The rate of responding to the CS was marginally correlated with $\log(T)$ (see *Appendix 1—figure 2A*), $r = -0.60$, $p = .025$, but was not correlated with either $\log(C)$, $r = -0.38$, $p = .181$, or $\log(C/T)$, $r = -0.08$, $p = .778$.

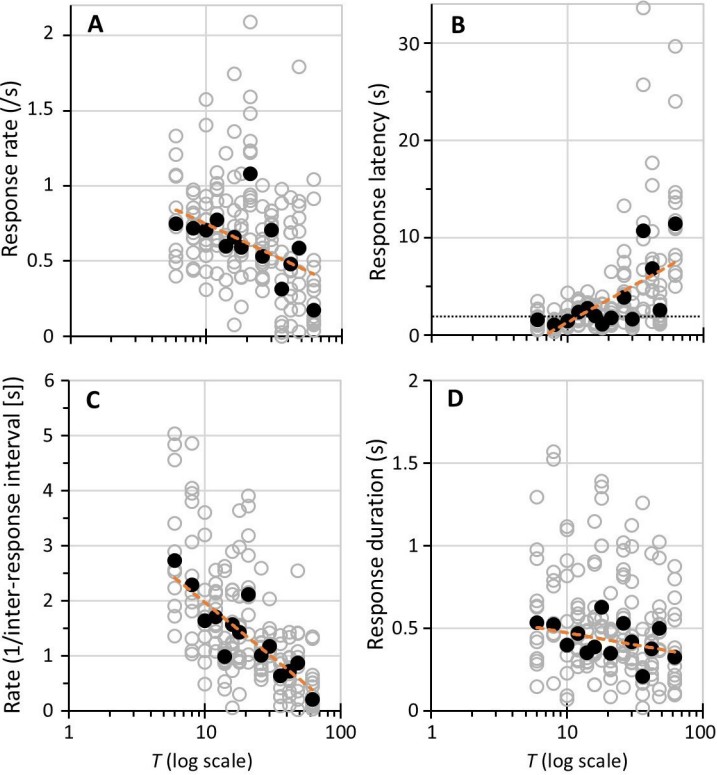

**Appendix 1—figure 2.** Response metrics, over the final 5 conditioning sessions, plotted against T. Open grey circles show data for individual rats. Black filled circles show mean data for each group. The dashed orange lines are the best-fitting regression lines.

More detailed analyses were conducted after segmenting the response rate into 3 separate components: (1) the latency to the first response in a trial; (2) the mean duration of each response (the time spent in the magazine); and (3) the inverse of the mean inter-response interval, 1/IRI, which equals the response rate after excluding the latency and response durations. The group medians for each of these indices was correlated with the log of $C$, $T$, and $C/T$. None correlated significantly with $\log(C)$, largest $r = -0.39$, $p = .168$, or with $\log(C/T)$, largest $r = -0.17$, $p = .557$. On the other hand, $\log(T)$ correlated significantly with latency, $r = 0.68$, $p = .007$ (*Appendix 1—figure 2B*), and with 1/IRI, $r = 0.87$, $p < .001$ (*Appendix 1—figure 2C*). Duration of responding did not correlate significantly with $\log(T)$, $r = -0.42$, $p = .149$ (*Appendix 1—figure 2D*). As is evident in *Appendix 1—figure 2B*, the positive correlation between latency and $\log(T)$ was largely confined to groups with long CS-US intervals ($T > 25$ s), whereas latency varied little among groups with shorter CS-US intervals. This invariance at short CS-US intervals may have been due to a floor effect because mean latencies to first response did not decrease below 2 s (horizontal dotted line in *Appendix 1—figure 2B*). This is also consistent with the plots of response by time-in-CS, shown in plot 15 of *Appendix 1—figure 1*, where responding to the CS was low for the first few seconds after CS onset. This apparent

floor effect might reflect a constraint on how quickly the rats can commence responding after CS onset or it might have arisen because 2 s was the minimum CS-US interval used in this experiment. Regardless, these analyses indicate that neither duration of responding nor latency to first response are good markers of what the rats learn about the rate of reinforcement of the CS. By contrast, the response rate, 1/IRI, varied systematically across the entire range of values of $T$ (**Appendix 1—figure 2C**). This confirms earlier evidence that rats' response rates scale with the log of the reinforcement rate (**Harris and Carpenter, 2011**).

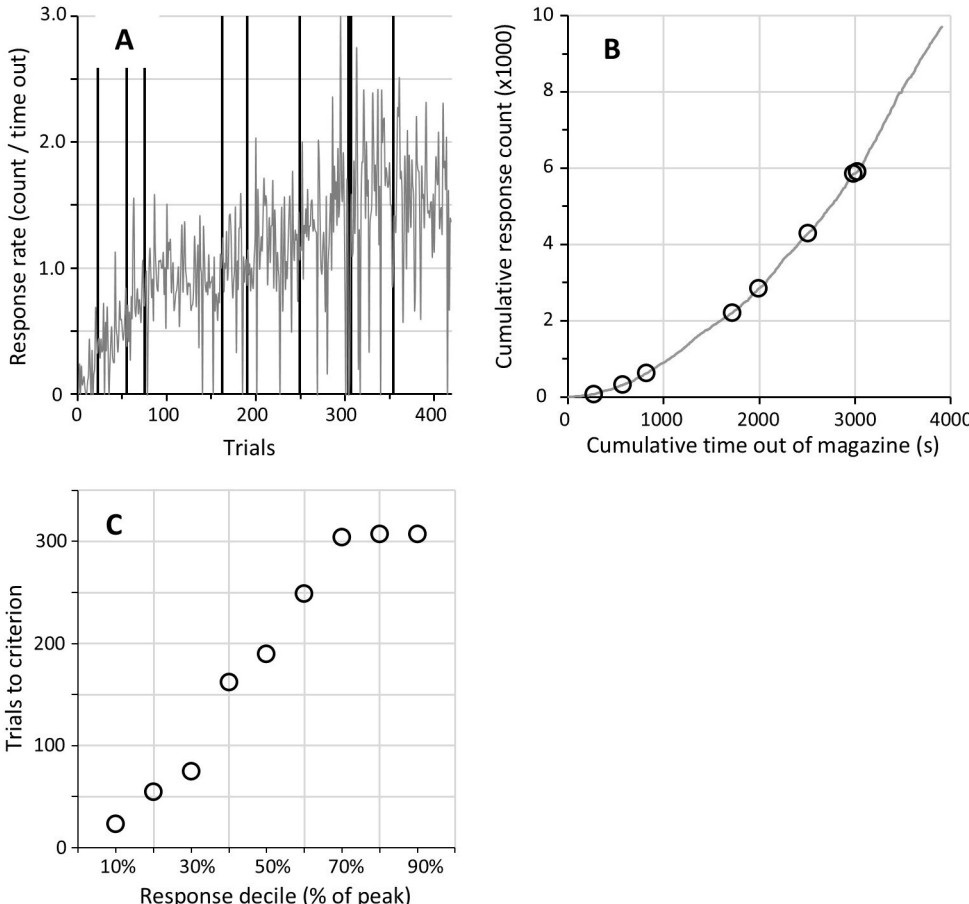

**Appendix 1—figure 3.** Response rate and trials to criterion for an individual rat. The grey line in A shows the response rate on each trial for an individual rat (Rat 30, Group C/T = 4.5). The response rate was calculated as the number of responses during the CS divided by the total time out of the magazine but excluding the latency to first response (i.e., the total time across which the animal could respond). In B, the cumulative response count across trials for the same rat is plotted against the cumulative opportunity to respond (cumulative time out of the magazine). The slope of this cumulative function was used to identify the trial on which the rat's response rate reached each decile (from 10% to 90%) of its peak response rate, as shown in C. These trials are also marked as circles on the response plots in A and B.

## Additional analyses of trials to criterion

In addition to the analysis we described in the main article, we also conducted analyses of trials to criterion on our data following two approaches used previously by other researchers. One of these methods was used by **Thrailkill et al., 2020**. Their method identified the point of acquisition using a criterion in which the mean response rate during the CS had to exceed the mean pre-CS response rate by at least 1.5 responses/10 s (9 responses/min) on 3 consecutive 4-trial blocks. We applied this method to our data but we multiplied the obtained value by 4 in order to convert the blocks-to-criterion results back into trials-to-criterion so that the results could be compared directly to those of our own analyses. The trials to criterion for each group is shown in **Appendix 1—figure 4**, plotted against the C/T ratio, which also plots for comparison the trials to criterion identified

as the trial after which the $nD_{kl}$ became permanently positive (shown in *Figure 3A* of the main article). The group means of trials to criterion produced using the two methods correlated well with each other, r = .85, *p* < .001. However, the values obtained using the Thrailkill et al. method were on average 11 times higher than those identified using the $nD_{kl}$, t(167) = 6.27, *p* < .001. This difference indicates that the criterion used by Thrailkill et al. was sensitive to more advanced stages of response acquisition. Nonetheless, the scores from this method did scale with the log of the *C/T* ratio, as shown in *Appendix 1—figure 4*. The same correlational analyses described in the main article using the $nD_{kl}$ criterion were conducted from the results using the Thrailkill et al. method. The correlation between log of trials to criterion and log(*C/T*) was significant, r = -0.83, *p* < .001, whereas the correlations with log(*T*) and log(*C*) fell short of the corrected level of significance, r = 0.61 and -0.61, *p* = .021 and .020, respectively. Given that log(*C/T*) was correlated with both log(*C*) and log(*T*), partial correlations were calculated to assess the relationship between each variable and trials to criterion. The correlation between log(*C/T*) and trials to criterion remained significant after partialling out the effect of log(*T*), r = -.80, *p* < .001, or log(*C*), r = -.80, *p* = .001.

The analyses described above show that the relationship between learning rate (trials to an acquisition criterion) and informativeness still hold when using the acquisition criterion adopted by *Thrailkill et al., 2020*. Nonetheless, that criterion identifies a later point in the learning process than the point identified using the $nD_{kl}$. Therefore, their method is more likely to be affected by factors, such as *T*, that affect the subsequent strength of responding. This may explain why Thrailkill et al. observed an effect of *T* on the rate of learning even when *C/T* was held constant, whereas our analysis, using the $nD_{kl}$, did not.

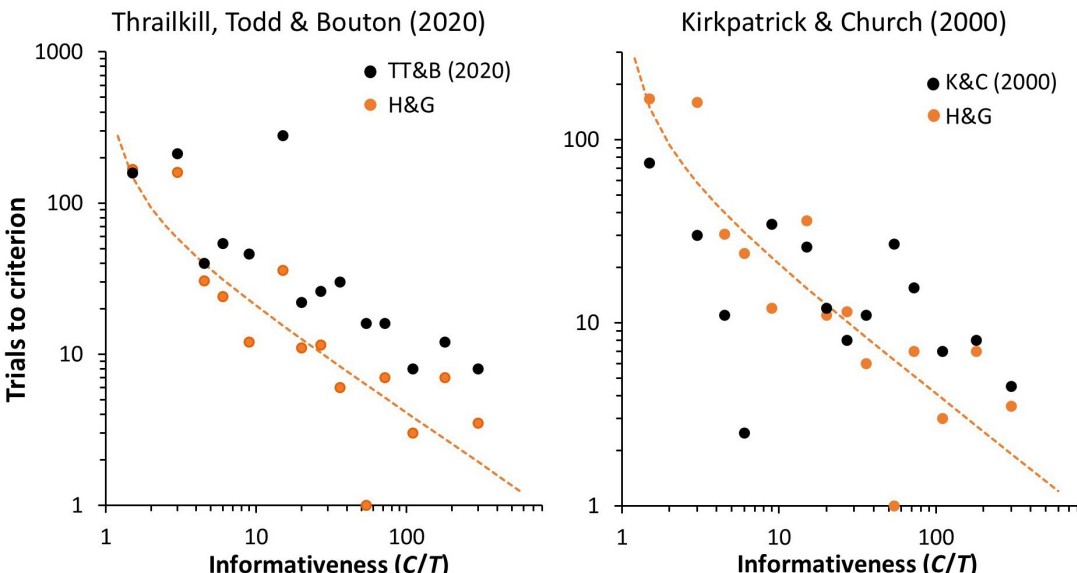

**Appendix 1—figure 4.** The mean number of trials to criterion for each of the 14 groups of rats (with different C/T ratios) in the current experiment. In the plot on the left, the filled black circles show the number of trials for each group to reach the criterion used by *Thrailkill et al., 2020*. In the plot on the right, the filled black circles show the number of trials for each group to reach the criterion used by *Kirkpatrick and Church, 2000*. For reference, the orange circles in both graphs show the number of trials for the difference between the cumulative CS response rate and cumulative ITI rate to become permanently positive (the criterion shown in *Figure 2A* of the main article).

The final method we have used to analyse our data followed the method described by *Kirkpatrick and Church, 2000*. This was based on a discrimination ratio calculated as the number of responses ($R_{CS}$) made during a brief time window (of length 2/15ths of *T*) in the middle of the CS-US interval divided by the same response count plus a baseline response count ($R_b$) measured during a window of equivalent length in the middle of the pre-CS interval (i.e., $R_{CS}/[R_{CS}+R_b]$). To identify how quickly responding to the CS was acquired, Kirkpatrick and Church fitted a low-high step function to the discrimination ratios across trials to identify the trial, *t*, at which there was maximal change in the ratio averaged across trials before *t* versus trials after *t*. We have applied Kirkpatrick and Church's step-function method to discrimination ratios calculated from our data but using response rates

across the full CS and pre-CS trial lengths rather than sampled from a small window (2/15$^{ths}$ of $T$) within those intervals. The trials to criterion identified with this method are shown in the right graph of *Appendix 1—figure 4*, plotted against the $C/T$ ratio. As is evident in the scatter of values shown in the figure, this method for estimating trials to acquisition was less sensitive to differences in $C/T$ than any of our criteria or the method used by *Thrailkill et al., 2020*. The log of trials to criterion was not significantly correlated with log($C/T$), r = -.50, $p$ = .071, or log($C$), r = -.35, $p$ = .221, or log($T$), r = .40, $p$ = .157. The data obtained using Kirkpatrick and Church's method were not significantly correlated with the data produced by the criterion we used, r = .409, $p$ = .147, and fell short of a significant correlation with the data produced using Thrailkill et al.'s method, r = .57, $p$ = .034. One limitation with the Kirkpatrick and Church measure is that simple changes in response rates between the CS and baseline undergo a non-linear transformation when computed as a ratio. For example, as the pre-CS response count goes to zero, the ratio goes to a maximum of 1 regardless of how many responses are made during the CS. Because this algorithm uses response rates across the whole experiment to identify the acquisition trial, the trial selected will depend on when the discrimination ratio reached its peak, which will itself be strongly influenced by the decline in baseline responding, $R_b$.

