## [Editor Report · eLife Assessment]

This paper presents **fundamental** research showing that the acquisition and expression of Pavlovian conditioned responding are lawfully related to temporal characteristics of an animal's conditioning experience. It showcases a rigorous experimental design, several different approaches to data analysis, careful consideration of prior literature, and a thorough introduction. The evidence supporting the conclusions is **compelling**. The paper will have a general appeal to those interested in the behavioral and neural analysis of Pavlovian conditioning.

---

## [Referee Report · Reviewer #1 (Public review)]

The authors did a good job addressing my concerns. This is an important paper on the basis of appetitive associative learning.

---

## [Referee Report · Reviewer #2 (Public review)]

A long-standing debate in the field of Pavlovian learning relates to the phenomenon of timescale invariance in learning i.e. that the rate at which an animal learns about a Pavlovian CS is driven by the relative rate of reinforcement of the cue (CS) to the background rate of reinforcement. In practice, if a CS is reinforced on every trial, then the rate of acquisition is determined by the relative duration of the CS (T) and the ITI (C = inter-US-interval = duration of CS + ITI), specifically the ratio of C/T. Therefore, the point of acquisition should be the same with a 10s CS and a 90s ITI (T = 10; C = 90 + 10 = 100, C/T = 100/10 = 10) and with a 100s CS and a 900s ITI (T = 100; C = 900 + 100 = 1000, C/T = 1000/100 = 10). That is to say, the rate of acquisition is invariant to the absolute timescale as long as this ratio is the same. This idea has many other consequences, but is also notably different from more popular prediction-error based associative learning models such as the Rescorla-Wagner model. The initial demonstrations that the ratio C/T predicts the point of acquisition across a wide range of parameters (both within and across multiple studies) was conducted in Pigeons using a Pavlovian autoshaping procedure. What has remained under contention is whether or not this relationship holds across species, particularly in the standard appetitive Pavlovian conditioning paradigms used in rodents. The results from rodent studies aimed at testing this have been mixed, and often the debate around the source of these inconsistent results focuses on the different statistical methods used to identify the point of acquisition for the highly variable trial-by-trial responses at the level of individual animals.

The authors successfully replicate the same effect found in pigeon autoshaping paradigms decades ago (with almost identical model parameters) in a standard Pavlovian appetitive paradigm in rats. They achieve this through a clever change the experimental design, using a convincingly wide range of parameters across 14 groups of rats, and by a thorough and meticulous analysis of these data. It is also interesting to note that the two authors have published on opposing sides of this debate for many years, and as a result have developed and refined many of the ideas in this manuscript through this process.

Main findings

(1) The present findings demonstrate that the point of initial acquisition of responding is predicted by the C/T ratio.

(2) The terminal rates of responding to the CS appear to be related to the reinforcement rate of the CS (T; specifically, 1/T) but not its relation to the reinforcement rate of the context (i.e. C or C/T). In the present experiment, all CS trials were reinforced so it is also the case that the terminal rate of responding was related to the duration of the CS.

(3) An unexpected finding was that responding during the ITI was similarly related to the rate of contextual reinforcement (1/C). This novel finding suggests that the terminal rate of responding during the ITI and the CS are related to their corresponding rates of reinforcement. This finding is surprising as it suggests that responding during the ITI is not being driven by the probability of reinforcement during the ITI.

(4) Finally, the authors characterised the nature of increased responding from the point of initial acquisition until responding peaks at a maximum. Their analyses suggest that nature of this increase was best described as linear in the majority of rats, as opposed to the non-linear increase that might be predicted by prediction error learning models (e.g. Rescorla-Wagner). However, more detailed analyses revealed that these changes can be quite variable across rats, and more variable when the CS had lower informativeness (defined as C/T).

Strengths and Weaknesses:

There is an inherent paradox regarding the consistency of the acquisition data from Gibbon & Balsam's (1981) meta-analysis of autoshaping in pigeons, and the present results in magazine response frequency in rats. This consistency is remarkable and impressive, and is suggestive of a relatively conserved or similar underlying learning principle. However, the consistency is also surprising given some significant differences in how these experiments were run. Some of these differences might reasonably be expected to lead to differences in how these different species respond. For example:

The autoshaping procedure commonly used in the pigeons from these data were pretrained to retrieve rewards from a grain hopper with an instrumental contingency between head entry into the hopper and grain availability. During Pavlovian training, pecking the key light also elicited an auditory click feedback stimulus, and when the grain hopper was made available, the hopper was also illuminated.

In the present experimental procedure, the rats were not given contextual exposure to the pellet reinforcers in the magazine (e.g. a magazine training session is typically found in similar rodent procedures). The Pavlovian CS was a cue light within the magazine itself.

These design features in the present rodent experiment are clearly intentional. Pretraining with the reinforcer in the testing chambers would reasonably alter the background rate of reinforcement (parameter), so it make sense not to include this but differs from the paradigm used in pigeons. Having the CS inside the magazine where pellets are delivered provides an effective way to reduce any potential response competition between CS and US directed responding and combines these all into the same physical response. This makes the magazine approach response more like the pecking of the light stimulus in the pigeon autoshaping paradigm. However, the location of the CS and US is separated in pigeon autoshaping, raising questions about why the findings across species are consistent despite these differences.

Intriguingly, when the insertion of a lever is used as a Pavlovian cue in rodent studies, CS directed responding (sign-tracking) often develops over training such that eventually all animals bias their responding towards the lever than towards the US (goal-tracking at the magazine). However, the nature of this shift highlights the important point that these CS and US directed responses can be quite distinct physically as well as psychologically. Therefore, by conflating the development of these different forms of responding, it is not clear whether the relationship between C/T and the acquisition of responding describes the sum of all Pavlovian responding or predominantly CS or US directed responding.

Another interesting aspect of these findings is that there is a large amount of variability that scales inversely with C/T. A potential account of the source of this variability is related to the absence of preexposure to the reward pellets. This is normally done within the animals' homecage as a form of preexposure to reduce neophobia. If some rats take longer to notice and then approach and finally consume the reward pellets in the magazine, the impact of this would systematically differ depending on the length of the ITI. For animals presented with relatively short CSs and ITIs, they may essentially miss the first couple of trials and/or attribute uneaten pellets accumulating in the magazine to the background/contextual rate of reinforcement. What is not currently clear is whether this was accounted for in some way by confirming when the rats first started retrieving and consuming the rewards from the magazine.

While the generality of these findings across species is impressive, the very specific set of parameters employed to generate these data raise questions about the generality of these findings across other standard Pavlovian conditioning parameters. While this is obviously beyond the scope of the present experiment, it is important to consider that the present study explored a situation with 100% reinforcement on every trial, with a variable duration CS (drawn form a uniform distribution), with a single relatively brief CS (maximum of 122s) CS and a single US. Again, the choice of these parameters in the present experiment is appropriate and very deliberately based on refinements from many previous studies from the authors. This includes a number of criteria used to define magazine response frequency which includes discarding specific responses (discussed and reasonably justified clearly in the methods section). Similarly, the finding that terminal rates of responding are reliably related to 1/T is surprising, and it is not clear whether this might be a property specific to this form of variable duration CS, the use of a uniform sampling distribution, or the use of only a single CS. However, it is important to keeps these limitations in mind when considering some of the claims made in the discussion section of this manuscript that go beyond what these data can support.

---

## [Author Response]

The following is the authors’ response to the previous reviews.

**Recommendations for the authors:**

**Reviewer #1 (Recommendations for the authors):**
Conceptually, I feel that the authors addressed many concerns. However, I am still not convinced that their data support the strength of their claims. Additionally, I spent considerable time investigating the now freely available code and data and found several inconsistencies that would be critical to rectify. My comments are split into two parts, reflecting concerns related to the responses/methods and concerns resulting from investigation of the provided code/data. The former is described in the public review above. Because I show several figures to illustrate some key points for the latter part, an attached file will provide the second part: https://elife-rp.msubmit.net/elife-rp_files/2025/02/24/00136468/01/136468_1_attach_15_2451_convrt.pdf(1) This point is discussed in more detail in the attached file, but there are some important details regarding the identification of the learned trial that require more clarification. For instance, isn’t the original criterion by Gibbon et al. (1977) the first “sequence of three out of four trials in a row with at least one response”? The authors’ provided code for the Wilcoxon signed rank test and nDkl thresholds looks for a permanent exceeding of the threshold. So, I am not yet convinced that the approaches used here and in prior papers are directly comparable.

We agree that there remain unresolved issues with our two attempts to create criteria that match that used by Gibbon and Balsam for trials to criterion. Therefore, we have decided to remove those analyses and return to our original approach showing trials to acquisition using several different criteria so as to demonstrate that the essential feature of the results—the scaling between learning rate and information—is robust. Figure 2A shows the results for a criterion that identifies the trial after which the cumulative response rate during the CS (=cumulative CS response count from Trial 1 divided by cumulative CS time from Trial 1) is consistently above the cumulative overall response rate across the trial (i.e., including both the CS and ITI). These data compare the CS response rate with the overall response rate, rather than with ITI rate as done in the previous version (in Figure 3A of that submission), to be consistent with the subsequent comparisons that are made using the nDkl. (The nDkl relies on the comparison between the CS rate and the overall rate, rather than between the CS and ITI rates.) Figures 2B and 2C show trials to acquisition when two statistical criteria, based on the nDkl, are applied to the difference between CS and overall response rates (the criteria are for odds >= 4:1 and p<.05). As we now explain in the text, a statistical threshold is useful inasmuch as it provides some confidence to the claim that the animals had learned by a given trial. However, this trial is very likely to be after the point when they had learned because accumulating statistical evidence of a difference necessarily adds trials.

Also, there’s still no regression line fitted to their data (Fig 3’s black line is from Fig 1,according to the legends). Accordingly, I think the claim in the second paragraph of the Discussion that the old data and their data are explained by a model with “essentially the same parameter value” is not yet convincing without actually reporting the parameters of the regression. Related to this, the regression for their data based on my analysis appears to have a slope closer to -0.6, which does not support strict timescale invariance. I think that this point should be discussed as a caveat in the manuscript.

We now include regression lines fitted to our data in Figures 2A-C, and their slopes are reported in the figure note. We also note on page 14 of the revision that these regressions fitted to our data diverge from the black regression line (slope -1) as the informativeness increases. On pages 14-15, we offer an explanation for this divergence; that, in groups with high informativeness, the effective informativeness is likely to be lower than the assigned value because the rats had not been magazine trained which means they would not have discovered the food pellet as soon as it was released on the first few trials. On pages 15-16, we go on to note that evidence for a change in response rate during the CS in those very first few trials may have been missed because the initial response rates were very low in rats trained with very long inter-reinforcement intervals (and thus high informativeness). We also propose a solution to this problem of comparing between very low response rates, one that uses the nDkl to parse response rates into segments (clusters of trials with equivalent response rates). This analysis with parsed response rates provides evidence that differential responding to the CS may have been acquired earlier than is revealed using trial-by-trial comparisons.

(2) The authors report in the response that the basis for the apparent gradual/multiple step-like increases after initial learning remains unclear within their framework. This would be important to point out in the actual manuscript Further, the responses indicating the fact that there are some phenomena that are not captured by the current model would be important to state in the manuscript itself.

We have included a paragraph (on page 26) that discusses the interpretation of the steady/multi-step increase in responding across continued training.

(3) There are several mismatches between results shown in figures and those produced by the authors’ code, or other supplementary files. As one example, rat 3 results in Fig 11 and Supplementary Materials don’t match and neither version is reproduced by the authors’ code. There are more concerns like this, which are detailed in the attached review file.

Addressed next….

The following is the response to the points raised in Part 2 of Reviewer 1’s pdf.

(1a) I plotted the calculated nDkl with the provided code for rat 3 (Fig 11), but itlooks different, and the trials to acquisition also didn’t match with the table provided (average of ~20 trial difference). The authors should revise the provided code and plots. Further, even in their provided figures, if one compares rat 3 in Supplementary Materials to data from the same rat in Fig 11, the curves are different. It is critical to have reproducible results in the manuscript, including the ability to reproduce with the provided code.

We apologise for those inconsistencies. We have checked the code and the data in the figures to ensure they are all now consistent and match the full data in the nHT.mat file in OSF. Figures 11 and 12 from the previous version are now replaced with Figure 6 in the revised manuscript (still showing data from Rats 3 and 176). The data plotted in Fig 6 match what is plotted in the supplementary figures for those 2 rats (but with slightly different cropping of the x-axes) and all plots draw directly from nHT.mat.

(1b) I tried to replicate also Fig 3C with the results from the provided code, but I failed especially for nDkl > 2.2. Fig 3A and B look to be OK.

There was error in the previous Fig 3C which was plotting the data from the wrong column of the Trials2Acquisition Table. We suspect this arose because some changes to the file were not updated in Dropbox. However, that figure has changed (now Figure 2) as already mentioned, and no longer plots data obtained with that specific nDkl criterion. The figure now shows criteria that do not attempt to match the Gibbon and Balsam criterion.

(1c) The trials to learn from the code do match with those in the Trials2Acquisition Table, but the authors’ code doesn’t reproduce the reported trials to learn values in the nDkl Acquisition Table. The trials to learn from the code are ~20 trials different on average from the table’s ones, for 1:20, 1:100, and 1:1000 nDkl.

We agree that discrepancies between those different files were a source of potential confusion because they were using different criteria or different ways of measuring response rate (i.e., the “conventional” calculation of rate as number of responses/time, vs our adjusted calculation in which the 1^st^ response in the CS was excluded as well as the time spent in the magazine, vs parsed response rates based on inter-response intervals). To avoid this, there is now a single table called Acquisition_Table.xlsx in OSF that includes Trials to acquisition for each rat based on a range of criteria or estimates of response rate in labelled columns. The data shown in Figure 2 are all based on the conventional calculation of response rate (provided in Columns E to H of Acquisition_Table.xlsx). To make the source of these data explicit, we have provided in OSF the matlab code that draws the data from the nHT.mat file to obtain these values for trials-to-acquisition.

(1d) The nDkl Acquisition Table has columns with the value of the nDkl statistics at various acquisition landmarks, but the value does not look to be true, especially for rat 19. The nDkl curve provided by the authors (Supplementary Materials) doesn’t match the values in the table. The curve is below 10 until at least 300 trials, while the table reports a value higher than 20 (24.86) at the earliest evidence of learning (~120 trials?).

We are very grateful to the reviewer for finding this discrepancy in our previous files. The individual plots in the Supplementary Materials now contain a plot of the nDkl computed using the conventional calculation of response rate (plot 3 in each 6-panel figure) and a plot of the nDkl computed using the new adjusted calculation of response rate (plot 4). These correspond to the signed nDkl columns for each rat in the full data file nHT.mat. The nDkl values at different acquisition landmarks included in Acquisition_Table.xlsx (Cols AB to AF) correspond to the second of these nDkl formulations. We point out that, of the acquisition landmarks based on the conventional calculation of response rate (Cols E to J of Acquisition_Tabls.xlsx), only the first two landmarks (CSrate>Contextrate and min_nDkl) match the permanently positive and minimum values of the plotted nDkl values. This is because the subsequent acquisition landmarks are based on a recalculation of the nDkl starting from the trial when CSrate>ContextRate, whereas the plotted nDkl starts from Trial 1.

(2) The cumulative number of responses during the trial (Total) in the raw data table is not measured directly, but indirectly estimated from the pre-CS period, as (cumNR_Pre*[cumITI/cumT_Pre])+ cumNR_CS (cumNR_Pre: cumulative nose-poke response number during pre-CS period; cumITI: cumulative sum of ITI duration; cumT_Pre: cumulative pre-CS duration; cumNR_CS: cumulative response number during CS), according to ‘Explanation of TbyTdataTable (MATLAB).docx’.Why not use the actual cumulative responses during the whole trial instead of using a noisier measure during a smaller time window and then scaling it for the total period?

Unfortunately, the bespoke software used to control the experimental events and record the magazine activity did not record data continuously throughout the experiment. The ITI responses were only sampled during a specified time-window (the “pre-CS” period) immediately before each CS onset. Therefore, response counts across the whole ITI had to be extrapolated.

(3) Regarding the “Matlab code for Find Trials to Criterion.docx”:(a) What’s the rationale for not using all the trials to calculate nDkl but starting the cumulative summation from the earliest evidence trial (truncated)? Also, this procedure is not described in the manuscript, and this should be mentioned.

The procedure was perhaps not described clearly enough in the previous manuscript. We have expanded that text to make it clearer (page 12) which includes the text…

“We started from this trial, rather than from Trial 1, because response rate data from trials prior to the point of acquisition would dilute the evidence for a statistically significant difference in responding once it had emerged, and thereby increase the number of trials required to observe significant responding to the CS. The data from Rat 1 illustrates this point. The CS response rate of Rat 1 permanently exceeded its overall response rate on Trial 52 (when the nD_KL_ also became permanently positive). The nD_KL_, calculated from that trial onwards, surpassed 0.82 (odds 4:1) after a further 11 trials (on Trial 63) and reached 1.92 (p < .05) on Trial 81. By contrast, the nD_KL_ for this rat, calculated from Trial 1, did not permanently exceed 0.82 until Trial 83 and did not exceed 1.92 until Trial 93, adding 10 or 20 trials to the point of acquisition.”

(3b) The authors' threshold is the trial when the nDkl value exceeds the threshold permanently. What about using just the first pass after the minimum?

Rat 19 provides one example where the nDkl was initially positive, and even exceeded threshold for odds 4:1 and p<.05, but was followed by an extended period when the nDkl was negative because the CS response rate was less than the overall response rate. It illustrates why the first trial on which the nDkl passes a threshold cannot be used as a reliably index of acquisition.

(3c) Can the authors explain why a value of 0.5 is added to the cumulative response number before dividing it by the cumulative time?

This was done to provide an “unbiased” estimate of the response count because responses are integers. For example, if a rat has made 10 responses over 100 s of cumulative CS time, the estimated rate should be at least 10/100 but could be anything up to, but not including, 11/100. A rate of 10.5/100 is the unbiased estimate. However, we have now removed this step when calculating the nDkl to identify trials to acquisition because we recognise that it would represent a larger correction to the rate calculated across short intervals than across long intervals and therefore bias comparison between CS and overall response rates that involve very different time durations. As such, the correction would artefactually inflate evidence that the CS response rate was higher than the contextual response rate. However, as noted earlier in this reply, we have now instituted a similar correction when calculating the pre-CS response rate over the final 5 sessions for rats that did not register a single response (hence we set their response count to 0.5).

(3d) Although the authors explain that nDkl was set to negative if pre-CS rate is higher than CS rate, this is not included in the code because the code calculates the nDkl using the truncated version, starting to accumulate the poke numbers and time from the earliest evidence, thus cumulative CS rate is always higher than cumulative contextual rate. I expect then that the cumulative CS rate will be always higher than the cumulative pre-CS rate.

Yes, that is correct. The negative sign is added to the nDkl when it is computed starting from Trial 1. But when it is computed starting from the trial when the CS rate is permanently > the overall rate, there is no need to add a sign because the divergence is always in the positive direction.

(3e) Regarding the Wilcoxon signed rank test, please clarify in the manuscript that the input ‘rate’ is not the cumulative rate as used for the earliest evidence. Please also clarify if the rates being compared for the signed nDkl are just the instantaneous rates or the cumulative ones. I believe that these are the ‘cumulative’ ones (not as for Wilcoxon signed rank test), because if not, the signed nDkl curve of rat 3 would fluctuate a lot across the x-axis.

The reviewer is correct in both cases. However, as already mentioned, we have removed the analysis involving the Wilcoxon test. The description of the nDkl already specifies that this was done using the cumulative rates.

(4) Supplemental table ‘nDkl Acquisition Table.xlsx’ 3rd column (“Earliest”) descriptions are unclear.(a) It is described in the supplemental ‘Explanation of Excel Tables.docx’ as the ‘earliest estimate of the onset of a poke rate during the CSs higher than the contextual poke rate’, while the last paragraph of the manuscript’s method section says ‘Columns 4, 5 and 6 of the table give the trial after which conditioned responding appeared as estimated in the above described three different ways— by the location of the minimum in the nDkl, the last upward 0 crossings, and the CS parse consistently greater than the ITI parse, respectively. Column 3 in that table gives the minimum of the three estimates.’ I plotted the data from column 3 (right) and comparing them with Fig 3A (left) makes it clear that there’s an issue in this column. If the description in the ‘Explanation of Excel Tables.docx’ is incorrect, please update it.

We agree that the naming of these criteria can cause confusion, hence we have changed them. On page 9 we have replaced “earliest” with “first” in describing the criterion plotted in Figure 2A showing the trial starting from which the cumulative CS response rate permanently exceeded the cumulative overall rate. What is labelled as “Earliest” in “Acquisition_Table.xlsx” is, as the explanation says, the minimum value across the 3 estimates in that table.

(b) Also, the term ‘contextual poke rate’ in the 3rd column’s description isconfusing as in the nDkl calculation it represents the poke rate during all the training time, while in the first paragraph of the ‘Data analysis’ part, the earliest evidence is calculated by comparing the ITI (pre-CS baseline) poke rate.

Yes, we have kept the term “contextual” response rate to refer to responding across the whole training interval (the ITI and the CS duration). This is used in calculation of the nDkl. For consistency with this comparison, we now take the first estimate of acquisition (in Fig 2A) based on a comparison between the CS rate and the overall (context) rate (not the pre-CS rate).

**Reviewer #2 (Recommendations for the authors):**
In response to the Rebuttal comments:Analytical (1) relating to Figure 3C/DThis is a reasonable set of alternative analyses, but it is not clear that it answers the original comment regarding why the fit was worse when using a theoretically derived measure. Indeed, Figure 3C now looks distinctly different to the original Gibbon and Balsam data in terms of the shape of the relationship (specifically, the Group Median - filled orange circles) diverge from the black regression line.

As mentioned in response to Reviewer 1, there was a mistake in Figure 3C of the revised manuscript. The figure was actually plotting data using a more stringent criterion of nDkl > 5.4, corresponding to p<0.001. The figure was referencing the data in column J of the public Trials2Acquisition Table. The data previously plotted in Figure 3C are no longer plotted because we no longer attempt to identify a criterion exactly matching that used by Gibbon and Balsam.

We agree that the data shown in the first 3 panels of Figure 2 do diverge somewhat from the black regression line at the highest levels of informativeness (C/T ratios > 70), and the regression lines fitted to the data have slopes greater than -1. We acknowledge this on page 14 of the revised manuscript. Since Gibbon and Balsam did not report data from groups with such high ratios, we can’t know whether their data too would have diverged from the regression line at this point. We now report in the text a regression fitted to the first 10 groups in our experiment, which have C/T ratios that coincide with those of Gibbon and Balsam, and those regression lines do have slopes much closer to -1 (and include -1 in the 95% confidence intervals). We believe the divergence in our data at the high C/T ratios may be due to the fact that our rats were not given magazine training before commencing training with the CS and food. Because of this, it is quite likely that many rats did not find the food immediately after delivery on the first few trials. Indeed, in subsequent experiments, when we have continued to record magazine entries after CS-offset, we have found that rats can take 90 s or more to enter the magazine after the first pellet delivery. This delay would substantially increase the effective CS-US interval, measured from CS onset to discovery of the food pellet by the rat, making the CS much less informative over those trials. We now make this point on pages 14-15 of the revised manuscript.

Analytical (2)We may have very different views on the statistical and scientific approaches here.This scalar relationship may only be uniquely applicable to the specific parameters of an experiment where CS and US responding are measured with the same behavioral response (magazine entry). As such, statements regarding the simplicity of the number of parameters in the model may simply reflect the niche experimental conditions required to generate data to fit the original hypotheses.

To the extent that our data are consistent with the data reported decades ago by Gibbon and Balsam indicates the scalar relationship they identified is not unique to certain niche conditions since those special conditions must be true of both the acquisition of sign-tracking responses in pigeons and magazine entry responses in rats. How broadly it applies will require further experimental work using different paradigms and different species to assess how the rate of acquisition is affected across a wide range of informativeness, just as we have done here.